# PhysioWave: A Multi-Scale Wavelet-Transformer for Physiological Signal Representation

**Yanlong Chen**[1]    **Mattia Orlandi**[2]    **Pierangelo Maria Rapa**[2]
**Simone Benatti**[3]    **Luca Benini**[1,2]    **Yawei Li**[1]*

[1]IIS, ETH Zurich
[2]DEI, University of Bologna
[3]DIEF, University of Modena and Reggio Emilia

yanlchen@student.ethz.ch
{mattia.orlandi, pierangelomaria.rapa, simone.benatti}@unibo.it
lbenini@iis.ee.ethz.ch, li.yawei.ai@gmail.com

## Abstract

Physiological signals are often corrupted by motion artifacts, baseline drift, and other low-SNR disturbances, which pose significant challenges for analysis. Additionally, these signals exhibit strong non-stationarity, with sharp peaks and abrupt changes that evolve continuously, making them difficult to represent using traditional time-domain or filtering methods. To address these issues, a novel wavelet-based approach for physiological signal analysis is presented, aiming to capture multi-scale time-frequency features in various physiological signals. Leveraging this technique, two large-scale pretrained models specific to EMG and ECG are introduced for the first time, achieving superior performance and setting new baselines in downstream tasks. Additionally, a unified multi-modal framework is constructed by integrating pretrained EEG model, where each modality is guided through its dedicated branch and fused via learnable weighted fusion. This design effectively addresses challenges such as low signal-to-noise ratio, high inter-subject variability, and device mismatch, outperforming existing methods on multi-modal tasks. The proposed wavelet-based architecture lays a solid foundation for analysis of diverse physiological signals, while the multi-modal design points to next-generation physiological signal processing with potential impact on wearable health monitoring, clinical diagnostics, and broader biomedical applications. ⚕ Code and data are available at: github.com/ForeverBlue816/PhysioWave

## 1    Introduction

Physiological signals such as electroencephalography (EEG), electromyography (EMG), and electrocardiography (ECG) are essential for health monitoring, clinical diagnosis, and brain–computer interfacing [1]. Although large-scale foundation models for generic time-series data have recently shown remarkable success [2, 3], pretrained networks for biosignals remain scarce. While several EEG-specific encoders have been developed, such as LaBraM, which enables cross-dataset learning by segmenting signals into channel patches and training a vector-quantized neural spectrum tokenizer [4], and EEGPT, a pretrained Transformer designed for universal EEG feature extraction that combines masked reconstruction with spatio-temporal representation alignment to mitigate issues of low SNR and inter-subject variability [5], similar models for other biosignals are still lacking.

---

*Corresponding author

39th Conference on Neural Information Processing Systems (NeurIPS 2025).

Biosignals differ fundamentally from images or language: they evolve continuously, exhibit strong non-stationarity, and are often corrupted by motion artifacts, baseline drift, and other low-SNR disturbances [6]. These challenges, compounded by high inter-subject variability, hinder the application of standard deep learning techniques [7]. As a result, there is an urgent need for modeling frameworks that can naturally accommodate the multi-scale, noisy, and heterogeneous nature of physiological time-series.

Traditional time-domain methods often overlook the rich spectral content of biosignals [8], while frequency-domain methods like fixed-window Fourier transforms assume stationarity, losing temporal resolution in rapidly changing signals [9]. In contrast, *wavelet-based* time-frequency methods can adaptively capture both fast transients and slower dynamics, making them ideal for nonstationary physiological data [10, 11]. However, modalities such as optical biosensing and EEG differ significantly in terms of dimension, sampling rate, and resolution, requiring modality-specific preprocessing. Existing segmentation and tokenization pipelines are not cross-modal, making them unsuitable for handling diverse signal types [12]. To address these issues, we propose a learnable wavelet decomposition pipeline that automatically selects and fuses multi-resolution filters based on signal content. Instead of relying on hand-engineered wavelet families, our method uses an Adaptive Wavelet Selector to assign optimal wavelet kernels for each channel, then aggregates approximate and detail components across multiple scales. This approach effectively captures both transient spikes (e.g., EMG bursts or ECG QRS complexes) and sustained low-frequency fluctuations, providing a robust front-end for downstream tasks [13, 14].

Although self-supervised learning has demonstrated success in time-series data, methods inspired by natural language processing often discard contiguous tokens in the time domain [2, 15]. Unlike words, raw biosignal segments do not neatly correspond to meaningful units, and randomly discarding them may remove key events or mask redundant portions, ultimately limiting model performance [16]. To overcome this, we introduce *Frequency-guided Masking (FgM)*, a mechanism that selectively occludes the most informative segments. Inspired by SpecAugment's frequency masking and the masked acoustic modeling in HuBERT [17, 18], FgM measures the spectral energy of each patch to identify high-information regions. By masking high-energy patches more frequently, FgM forces the model to infer crucial details from context, thereby tightening the information bottleneck. A mix of energy-driven and random masking ensures training variability, preventing overfitting to fixed patterns and improving task generalization.

The wavelet-based decomposition and frequency-guided masking (FgM) work in tandem: adaptive wavelet filters separate the signal into multi-resolution bands, while FgM selectively masks bands with higher spectral energy, making them more likely to be occluded. This strategy of multiscale decomposition combined with energy-focused masking encourages the model to capture physiologically relevant patterns and reduce redundancy across frequencies. Empirically, masking high-energy bands yields richer, more discriminative features compared to random time-based masking, driving state-of-the-art performance in tasks such as arrhythmia detection (**66.7% F1 score** on PTB-XL [19]) and muscle activity classification (**94.5% accuracy** on EPN-612 [20], see Section 2.4).

Multi-modal biosignal learning faces two key challenges: first, extreme heterogeneity across modalities like EEG, EMG, and ECG, which operate at different sampling rates and temporal scales, leading to potential misalignment and spectral aliasing [21]; second, variability in signal quality due to motion artifacts or electrode drift [22]. To overcome these, we use modality-specific backbones: a pretrained encoder for EEG and newly pretrained PhysioWave encoders for EMG and ECG. These backbones remain frozen during downstream training, with only lightweight classification heads and fusion coefficients being learned. The fusion layer dynamically adjusts weights to prioritize reliable modalities, ensuring robust predictions. This linear-probing approach consistently outperforms single-modality baselines across multi-modal tasks (see Figure 5, including a **7.3% gain** in classification accuracy on DEAP [23], showcasing the effectiveness of dynamic, reliability-aware fusion for heterogeneous physiological data.

**In summary,** our work leverages wavelet-based decomposition and a FgM strategy to advance self-supervised learning in physiological signal processing. Specifically:

1. **PhysioWave: A versatile wavelet-driven architecture for physiological signals.** We propose PhysioWave, a versatile model framework applicable to diverse physiological signals, and using this architecture which accommodates physiological signals with different sampling rates and dimensions through a learnable wavelet decomposition pipeline and a

unified Transformer backbone, reducing the dependency on modality-specific preprocessing. A FgM mechanism is introduced to selectively occlude the most informative input segments, enhancing self-supervised learning.

2. **Enhanced representation learning.** Large-scale pre-trained models for EMG and ECG have been developed, trained on 823 GB of EMG data and 182 GB of ECG data, respectively, addressing the long-standing gap in foundational models for these modalities. Extensive evaluations across various tasks and datasets within each physiological modality demonstrate that PhysioWave consistently achieves state-of-the-art performance, underscoring its broad applicability and robustness (See Section 2.4).

3. **Unified multimodal framework.** By integrating our own pre-trained EMG and ECG models, which were developed using the PhysioWave architecture, with existing pre-trained EEG encoders, we build a unified framework that synergistically processes and fuses multimodal signals, producing superior performance compared to single-modal approaches.

# 2 Method

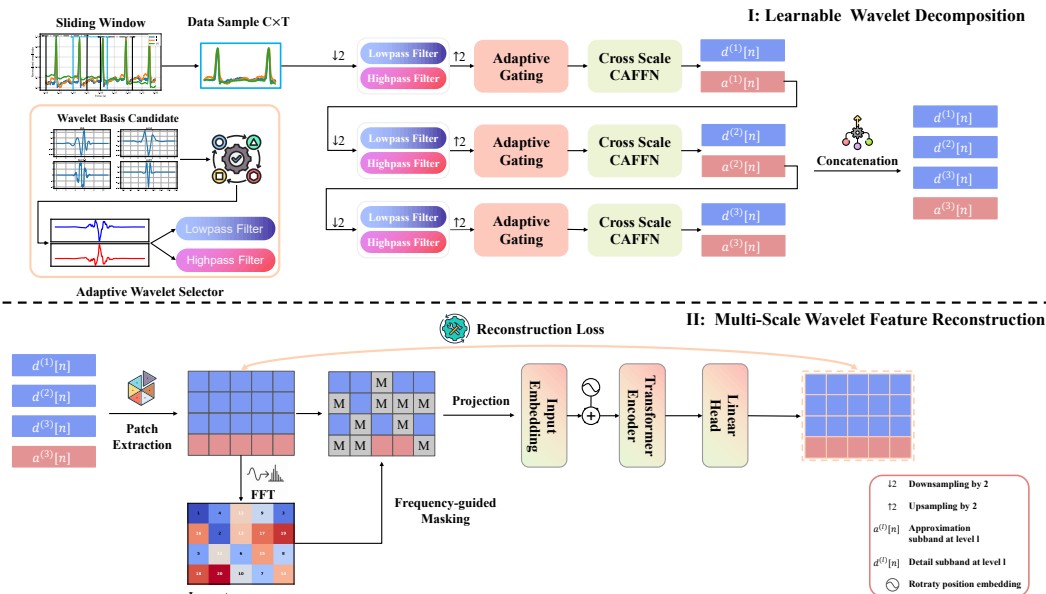

Figure 1: Model pretraining pipeline. The pipeline begins by initializing a set of standard wavelet functions (e.g., 'db6', 'sym4'), from which learnable low-pass and high-pass filters are generated. These filters are then used for wavelet decomposition to obtain multi-scale frequency-band representations. The decomposed features are processed into spatio-temporal patches, with importance scores computed using FFT-based spectral energy. High-scoring patches are masked and passed through Transformer layers, followed by a lightweight decoder for patch reconstruction.

**Model Architecture:** Figure 1 illustrates the end-to-end model pretraining pipeline. The raw multi-channel signal is first segmented into overlapping windows to form samples $x \in \mathbb{R}^{C \times T}$, where $C$ denotes the number of channels and $T$ the number of time steps. Each sample undergoes a learnable wavelet decomposition across $L$ levels, yielding multi-scale frequency-band representations: detail subbands $d^{(l)}[n] \in \mathbb{R}^{C \times T}$ for $l = 1, \ldots, L$ and a final approximation subband $a^{(L)}[n] \in \mathbb{R}^{C \times T}$. Concatenation of these $L + 1$ subband outputs along the channel axis produces a feature map $\mathrm{Spec}(X) \in \mathbb{R}^{((L+1)\,C) \times T}$ (Section 2.1). Next, the feature map is partitioned into uniform spatio-temporal patches: for each patch we compute its FFT-based spectral energy, blend this value with random noise to produce an importance score, and mask the highest-scoring patches by replacing their embeddings with a learnable <MASK> token. The remaining patches are projected into token embeddings and augmented with rotary positional embeddings (Section 2.2). Finally, the complete token sequence is passed through a stack of Transformer encoder layers, and a lightweight decoder

reconstructs the masked patches (Section 2.3). In addition, end-to-end fine-tuning is performed downstream for single-modality tasks, while in multimodal settings modality-specific predictions are aggregated to yield the final output (Section 2.4).

## 2.1 Learnable Wavelet Decomposition

This module decomposes a multichannel time series $\{X_c(t)\}_{c=1}^C$ of length $T$ into $L+1$ subbands per channel by iterating Analysis, Adaptive Gating, and Feature Fusion.

**Adaptive Wavelet Selector.** Low–pass and high–pass filter taps $h^{\mathrm{lo}}, h^{\mathrm{hi}} \in \mathbb{R}^{K_0}$, where $K_0$ is the original wavelet filter length are extracted from a chosen discrete wavelet using PyWavelets [24]. These taps are then resampled to length $K$ and normalized:

$$\tilde{h}^p(u) = \mathrm{Interp}(h^p, K)[u], \quad \tilde{h}^p \leftarrow \tilde{h}^p \frac{\|h^p\|_1}{\|\tilde{h}^p\|_1}, \quad p \in \{\mathrm{lo}, \mathrm{hi}\} \tag{1}$$

Each channel's depthwise filters are then initialized by copying these taps:

$$k_c^{\mathrm{low}}(u) := \tilde{h}^{\mathrm{lo}}(u), \quad k_c^{\mathrm{high}}(u) := \tilde{h}^{\mathrm{hi}}(u), \quad \forall c = 1, \ldots, C. \tag{2}$$

As depicted in the top-left corner of Figure 1, the model maintains $M$ candidate wavelet bases $\{(k_w^{\mathrm{low}}, k_w^{\mathrm{high}})\}_{w=1}^M$ to accommodate diverse signal characteristics. For each input $x \in \mathbb{R}^{C \times T}$, temporal information is aggregated via average pooling and the resulting feature vector is passed through a compact MLP to produce unnormalized selection scores. Applying a softmax yields selection weights

$$\alpha = \mathrm{Softmax}\big(\mathrm{MLP}(\mathrm{AvgPool}(x))\big) \in \mathbb{R}^M, \tag{3}$$

which are used to compute convex combinations of the candidate filters:

$$k^{\mathrm{low}} = \sum_{w=1}^M \alpha_w \, k_w^{\mathrm{low}}, \qquad k^{\mathrm{high}} = \sum_{w=1}^M \alpha_w \, k_w^{\mathrm{high}}. \tag{4}$$

These aggregated filters serve as the effective low- and high-pass filters for all subsequent analysis and downsampling operations, and during training their weights continue to be updated to adapt to the actual signal distribution.

**Analysis and Soft Gating.** The learnable wavelet front-end performs a multi-resolution analysis in which each stage halves the temporal resolution while preserving both low- and high-frequency content. Let downsampling and nearest-neighbor upsampling by a factor of two be

$$\begin{aligned} (\downarrow_2 x)[n] &= x[2n], \quad n = 0, \ldots, \lfloor \tfrac{T}{2} \rfloor - 1, \\ (\uparrow_2 x)[n] &= x\lfloor \tfrac{n}{2} \rfloor, \quad n = 0, \ldots, T - 1. \end{aligned} \tag{5}$$

At the first level ($\ell = 0$) every channel $c$ is filtered and downsampled,

$$a_c^{(1)}[n] = \sum_{u=0}^{K-1} X_c(2n+u) \, k_c^{\mathrm{low}}[u], \quad d_c^{(1)}[n] = \sum_{u=0}^{K-1} X_c(2n+u) \, k_c^{\mathrm{high}}[u], \tag{6}$$

and the process recurses for $\ell = 1, \ldots, L - 1$:

$$a_c^{(\ell+1)}[n] = \sum_{u=0}^{K-1} a_c^{(\ell)}(2n+u) \, k_c^{\mathrm{low}}[u], \quad d_c^{(\ell+1)}[n] = \sum_{u=0}^{K-1} a_c^{(\ell)}(2n+u) \, k_c^{\mathrm{high}}[u]. \tag{7}$$

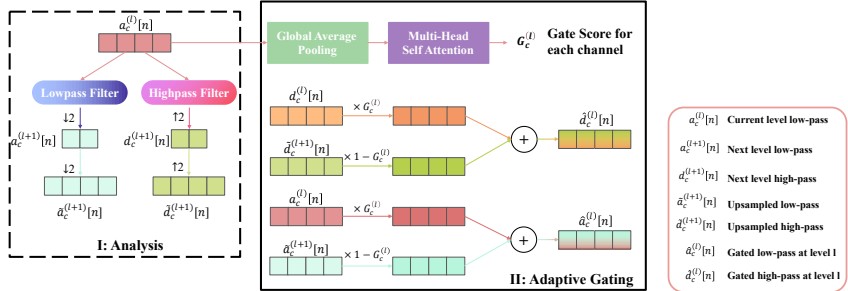

Figure 2: Analysis and soft gating process. The learnable wavelet front-end performs multi-resolution analysis by filtering and downsampling the input signal at each stage, preserving both low- and high-frequency components. At the first level ($\ell = 0$), the signal is decomposed into low-pass and high-pass components. This process recurses for $\ell = 1, \ldots, L-1$, applying downsampling at each level. After decomposition, the subbands are upsampled to the original resolution, and an adaptive gate $G_c^{(\ell)} \in [0,1]$ is learned for each channel using multi-head attention. The gate dynamically combines the original and upsampled signals, facilitating fine-scale detail insertion.

After each decomposition, the new subbands are upsampled back to the original length

$$\tilde{a}_c^{(\ell+1)}[n] = (\uparrow_2 a_c^{(\ell+1)})[n], \quad \tilde{d}_c^{(\ell+1)}[n] = (\uparrow_2 d_c^{(\ell+1)})[n], \tag{8}$$

so that fine-scale details can be re-inserted into the current resolution. Rather than performing a hard skip connection—as is common in U-Net–style architectures [25]—an adaptive gate $G_c^{(\ell)} \in [0,1]$ is estimated for every channel via multi-head attention pooling over $a_c^{(\ell)}$ (see Fig. 2). This gate weighs the contribution of the original and the upsampled signals,

$$\hat{a}_c^{(\ell)}[n] = G_c^{(\ell)}[n]\, a_c^{(\ell)}[n] + \left(1 - G_c^{(\ell)}[n]\right) \tilde{a}_c^{(\ell+1)}[n], \tag{9}$$

$$\hat{d}_c^{(\ell)}[n] = G_c^{(\ell)}[n]\, d_c^{(\ell)}[n] + \left(1 - G_c^{(\ell)}[n]\right) \tilde{d}_c^{(\ell+1)}[n]. \tag{10}$$

The soft-gating mechanism endows the model with three advantages: *(i)* it enables a learnable trade-off between current-level context and finer-scale detail, reducing aliasing and ringing artefacts that often arise from naive upsampling; *(ii)* the gate is estimated on a per-channel basis, allowing different physiological channels to emphasise frequency content most relevant to their noise characteristics; and *(iii)* by relying on attention rather than fixed skip connections, the model dynamically modulates information flow across levels, leading to more expressive multi-scale representations.

**Feature Fusion.** At each decomposition level $\ell = 1, \ldots, L$, the gated approximation–detail pair $[\hat{a}^{(\ell)}[n], \hat{d}^{(\ell)}[n]]$ is concatenated along the channel axis and reshaped into a $\mathbb{R}^{2C \times 1 \times T}$ feature map. This map is fed to a *Cross-Scale Channel-Aggregation Feed-Forward Network* (CAFFN; see Fig. 3) [26]. CAFFN first applies a lightweight channel-aggregation block and then performs multi-head attention where the current features act as queries and the flattened feature maps from all shallower levels provide keys and values. Formally, let $\mathbf{U}^{(\ell)}$ be the CAFFN output before cross-scale fusion; the refined representation is obtained as

$$\mathbf{Y}^{(\ell)}[n] = \mathbf{U}^{(\ell)}[n] + \beta\, \text{Attention}\!\left(\mathbf{U}^{(\ell)}[n], \{\mathbf{Y}^{(i)}[n]\}_{i<\ell}\right), \tag{11}$$

where the learnable scalar $\beta$ balances the current-level information with the context aggregated from coarser resolutions. This cross-scale fusion enables fine-grained subband features to be informed by long-range patterns captured at earlier levels, yielding scale-aware, frequency-aware representations for subsequent stages.

Finally, each refined map $\mathbf{Y}^{(\ell)}$ is split back into its approximation and detail halves, $\mathbf{Y}^{(\ell)} = [a^{(\ell)}, d^{(\ell)}]$, where $a^{(\ell)}, d^{(\ell)} \in \mathbb{R}^{C \times T}$. The multi-band representation is obtained by concatenating all detail subbands together with the final-level approximation:

$$\text{Spec}(X) = \left[d^{(1)}, d^{(2)}, \ldots, d^{(L)}, a^{(L)}\right] \in \mathbb{R}^{(L+1)C \times T}. \tag{12}$$

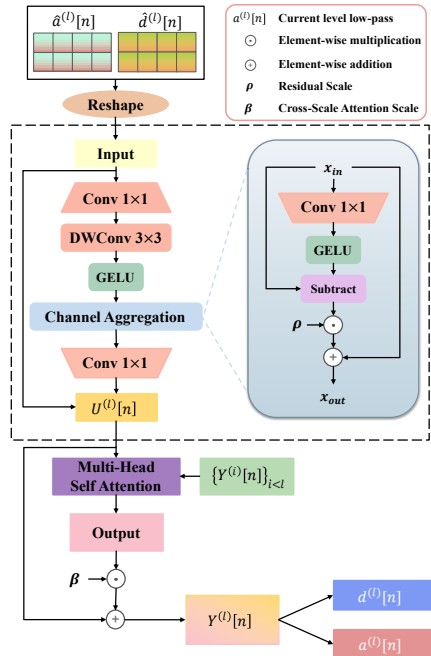

Figure 3: Cross-Scale CAFFN: This module refines multi-resolution features using convolution, channel aggregation, and self-attention.

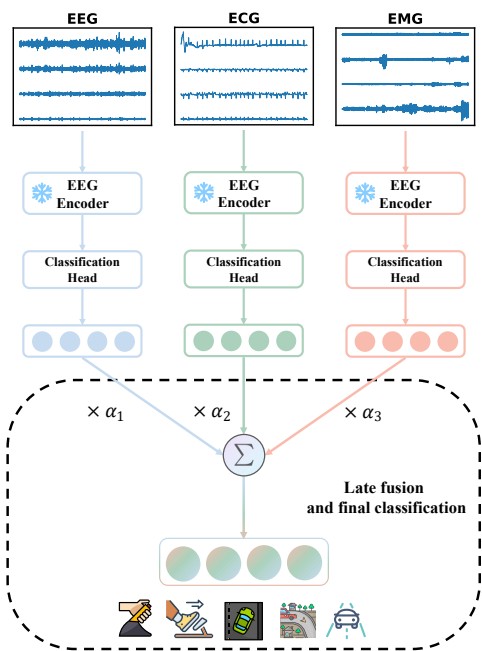

Figure 4: Multi-modal framework: Classification of driving behaviors in the MPDB dataset.

## 2.2 Frequency-guided Masking and Tokenisation

**Frequency-guided masking.** To address the challenge of selectively occluding informative segments, we introduce the frequency-guided masking process, as detailed in Algorithm 1. The multi-band feature map $\text{Spec}(X)$ from Eq. (12) is first sliced along the time axis into $N = \lfloor T/w \rfloor$ non-overlapping segments of length $w$, yielding a patch array of shape $\mathbb{R}^{N \times w}$ for each signal. In the next step, the FFT is applied to each patch to capture its frequency components. The energy of each patch in the frequency domain is then computed, which serves as an indicator of its importance. However, relying solely on frequency energy could lead to overfitting, especially in scenarios where certain high-energy patches are not necessarily the most informative. To mitigate this, random noise is introduced into the process. This randomness is controlled by the parameter $\alpha$, which governs the trade-off between the energy and the random noise [16]. The mask ratio, $\rho$, determines the proportion of patches to be masked. Together, the combination of frequency-guided masking and random noise ensures that the model learns to focus on the most informative parts of the signal, while also being robust to variations and missing information.

**Tokenisation.** Each masked or unmasked patch is projected to a $D$-dimensional token through a shared linear layer, forming the matrix $E \in \mathbb{R}^{N \times D}$. Rotary positional embeddings $\{\varsigma_n\}_{n=1}^{N} \subset \mathbb{R}^{D}$ are then added,

$$\tilde{E}_n = E_n + \varsigma_n, \qquad n = 1, \ldots, N. \tag{13}$$

The resulting sequence $\tilde{E} \in \mathbb{R}^{N \times D}$, together with the mask $M$, is forwarded to the Transformer encoder, which must reconstruct the masked, information-rich patches from context, thereby encouraging robust and frequency-aware representation learning.

## 2.3 Reconstruction

**Transformer encoder and lightweight decoder.** The encoder comprises $L$ *standard Transformer blocks* [27], where the only change is that the attention module uses *RoPE attention*. In RoPE attention the query and key sub-vectors of every head are rotated by deterministic sinusoidal factors that depend on their relative positions [28]. This rotation preserves dot-product magnitudes while

**Algorithm 1** Frequency-guided masking

---

**Require:** Signal $x \in \mathbb{R}^{C \times T}$, patch width $w$, mask ratio $\rho$, importance weight $\alpha$
1: $N \leftarrow \lfloor T/w \rfloor$            ▷ number of patches
2: Slice the time axis into $N$ non-overlapping patches $p_1, \dots, p_N \in \mathbb{R}^w$

---

3: **for** $n = 1$ **to** $N$ **do**            ▷ score each patch
4:      $F \leftarrow \mathrm{FFT}(p_n)$            ▷ frequency spectrum
5:      $e_n \leftarrow \sum_k |F_k|$            ▷ spectral energy
6: **end for**
7: Normalise $\mathbf{e} \leftarrow \dfrac{\mathbf{e} - \min(\mathbf{e})}{\max(\mathbf{e}) - \min(\mathbf{e})}$
8: Draw noise $z_n \sim \mathcal{U}(0,1)$ for $n = 1, \dots, N$
9: $s_n \leftarrow \alpha\, e_n + (1-\alpha)\, z_n$            ▷ blended score
10: $\mathbf{i} \leftarrow \mathrm{argsort}(\mathbf{s})$            ▷ ascending: low score first
11: $L \leftarrow \lfloor (1-\rho)N \rfloor$            ▷ patches to keep
12: keep $\leftarrow \mathbf{i}[1{:}L]$, mask $\leftarrow \mathbf{i}[L{:}N]$
13: Initialise $\mathbf{m} \leftarrow \mathbf{1}_N$;    $\mathbf{m}[\text{mask}] \leftarrow 0$            ▷ binary mask
14: Replace each $p_n$ with `<MASK>` if $m_n = 0$
15: Re-assemble the patched signal $\tilde{P} \in \mathbb{R}^{N \times w}$

---

16: **return** masked patches $\tilde{P}$, mask $\mathbf{m}$, sort indices $\mathbf{i}$

---

injecting position information, enabling each head to capture long-range temporal structure [29]. After the $L$ blocks the encoder outputs a latent sequence in which the masked tokens already carry context-inferred estimates [30, 31].

A deliberately shallow decoder then projects the latent width to $D_{\text{dec}}$, refines it with $L_{\text{dec}}$ vanilla Transformer blocks, and uses a final linear head to reconstruct every element of each patch, yielding $\hat{P} \in \mathbb{R}^{N \times w}$. Concentrating depth on the encoder while keeping the decoder light forces the model to store most semantic and spectral knowledge in the shared latent space [32].

**Patch-level reconstruction loss.** Let $\{\mathbf{p}_n\}_{n=1}^N$ and $\{\hat{\mathbf{p}}_n\}_{n=1}^N$ be the ground-truth and reconstructed *frequency-band patches* defined in Section 2.2, and let $\mathcal{M} = \{n \mid m_n = 1\}$ be the masked-patch index set. The model is trained to minimise the mean Smooth-L1 discrepancy over *only* the masked patches [33, 34]:

$$\mathcal{L} = \frac{1}{|\mathcal{M}|} \sum_{n \in \mathcal{M}} \mathrm{SmoothL1}(\hat{\mathbf{p}}_n, \mathbf{p}_n) \tag{14}$$

This objective 14 compels the network to recover the multiscale, frequency-band information concealed by the mask.

## 2.4 Methods for downstream tasks

**Single-modal setting.** For tasks that involve a single modality (EMG or ECG), the pretrained encoder is fine-tuned end-to-end. All patch tokens produced by the final encoder block are aggregated through mean pooling to obtain a global representation, which is then fed to a lightweight two-layer MLP to yield the final classification prediction.

**Multi-modal setting.** As illustrated in Figure 4, every pretrained encoder is kept *frozen* and only its dedicated classification head—along with a set of fusion coefficients—is trained. This *linear-probing* strategy preserves the representations learned during pretraining while mitigating the risk of over-fitting that often arises when a large-parameter model is adapted to a small downstream dataset [5]. Let $\mathcal{M}$ denote the set of modalities present in a particular experiment. Each modality $m \in \mathcal{M}$ produces logits $\mathbf{z}_m$, and the learnable fusion weights $\boldsymbol{\alpha} = \{\alpha_m\}_{m \in \mathcal{M}}$ are constrained by a softmax so that $\sum_{m \in \mathcal{M}} \alpha_m = 1$. The final prediction vector is obtained by

$$\mathbf{z}_{\text{fused}} = \sum_{m \in \mathcal{M}} \alpha_m\, \mathbf{z}_m \tag{15}$$

and $\mathbf{z}_{\text{fused}}$ is subsequently used to infer the task categories.

## 3 Experiments and Results

### 3.1 Datasets and Training Settings

Leveraging the **PhysioWave** architecture, two large-scale, modality-specific foundation models, **PhysioWave-ecg** and **PhysioWave-emg**, are pretrained on the most extensive open-access corpora currently available for their respective signal types (see Tables 15 and 16). PhysioWave-ecg is trained on approximately **182 GB** of twelve-lead ECG recordings, while PhysioWave-emg utilizes about **823 GB** of EMG data. For each modality, we provide three parameter configurations: *Small* (5M), *Base* (15M), and *Large* (37M). Both PhysioWave models share the same Transformer backbone; however, their learnable wavelet front-ends are *modality-aware* (see Tables 12 and 13 in Appendix C for full architectural and training details).

**Signal Preprocessing.** Each recording is denoised with a modality-specific band-pass filter followed by a 50 Hz notch. Traces with fewer than 12 ECG leads or 16 EMG electrodes are zero-padded and then resampled to 500 Hz (ECG) or 2 kHz (EMG). See Appendix D and E for full details.

**Downstream Tasks and Evaluation Protocol** The pretrained encoders are evaluated on the datasets listed in Table 17. All downstream experiments follow the single- and multi-modal procedures detailed in Section 2.4. Each benchmark is split *by subject* into **6:2:2** train/validation/test partitions to prevent subject leakage.

### 3.2 Downstream Experiment Results

**ECG Multi-label Classification Results.** Table 1 compares the proposed PhysioWave-ecg with recent large-scale pretrained ECG models. The *Small* model (5 M) already delivers competitive performance: on PTB-XL it raises F1 from 55.9 % (ECG-Chat [35], 13 B) to 65.8 % (+9.9 %), while attaining a comparable AUROC (92.7 % vs. 94.1%). Increasing capacity to *Base* and *Large* yields consistent gains; the 37 M variant sets new best scores on two of the three benchmarks, achieving **66.7 % / 94.6 %** (F1/AUROC) on PTB-XL and **54.8 % / 98.3 %** on Chapman–Shaoxing. On CPSC 2018 it surpasses the previous AUROC ceiling by +0.4 % (96.1 % vs. 95.7 %) and narrows the F1 gap to ECG-Chat from 80.1 % to 73.1 %.

Table 1: ECG rhythm classification results on three benchmark datasets.

| Method (year) | Params | PTB-XL | | CPSC 2018 | | Chapman-Shaoxing | |
|---|---|---|---|---|---|---|---|
| | | F1 | AUROC | F1 | AUROC | F1 | AUROC |
| ECG-Chat (2024) | 13B | 55.9 | 94.1 | **80.1** | 95.7 | — | — |
| MERL (2024) | 11M | 48.1 | 91.9 | 72.8 | 92.6 | — | 87.9 |
| MaeFE (2023) | 9M | 64.7 | 88.6 | 71.6 | 94.5 | — | — |
| OpenECG-SimCLR | 11M | 46.9 | 91.5 | 73.1 | 92.4 | 52.3 | 95.1 |
| OpenECG-BYOL | 11M | 47.7 | 91.1 | 72.8 | 92.6 | 51.5 | 94.8 |
| OpenECG-MAE | 11M | 48.1 | 90.9 | 74.5 | 93.2 | 50.8 | 94.2 |
| **Ours–Small** | 5M | 65.8 | 92.7 | 71.6 | 95.5 | 52.1 | 96.4 |
| **Ours–Base** | 15M | 64.5 | 93.4 | 72.5 | 95.9 | 53.8 | 97.2 |
| **Ours–Large** | 37M | **66.7** | **94.6** | 73.1 | **96.1** | **54.8** | **98.3** |

Note: Pink indicates the best results, blue indicates the second-best results.

Table 2 benchmarks the proposed *PhysioWave-EMG* against the only publicly released large-scale *generic* time-series models—*Moment* (385 M) and *OTiS* (45 M)—because no foundation model has yet been pretrained specifically for EMG signals [2, 36]. Even with just 5 M parameters, the *Small* variant already surpasses both baselines on the challenging EPN-612 dataset (93.1 % / 93.4 % versus

Table 2: Surface-EMG gesture recognition performance across three datasets.

| Method (year) | Params | NinaPro DB5 | | EPN-612 | | UCI EMG | |
|---|---|---|---|---|---|---|---|
| | | Acc. | F1 | Acc. | F1 | Acc. | F1 |
| Moment (2024) | 385M | 86.41 | 74.42 | 90.87 | 90.16 | 90.45 | 91.75 |
| OTiS (2024) | 45M | 85.31 | 72.61 | 87.55 | 88.03 | 90.62 | 89.28 |
| **Ours–Small** | 5M | 84.78 | 72.54 | 93.12 | 93.40 | 90.35 | 89.51 |
| **Ours–Base** | 15M | 86.02 | 73.78 | 93.68 | 93.91 | 91.92 | 92.77 |
| **Ours–Large** | 37M | **87.53** | **75.42** | **94.50** | **94.56** | **93.19** | **93.59** |

Note: Pink indicates the best results, blue indicates the second-best results.

90.9 % / 90.2 % for Moment). Scaling up the encoder brings steady gains: the 37 M *Large* model achieves new state-of-the-art results on all three benchmarks—NinaPro DB5 (+1.1 % accuracy over Moment), EPN-612 (+3.6 %), and UCI EMG (+2.6 % over OTiS)—while still using less than 1/10 of Moment's parameters.

**Multi-modal Classification Results.** The proposed multi-modal framework—built from the *Small* PhysioWave-ecg and PhysioWave-emg encoders together with an EEG branch (EEGPT for DEAP, LaBraM for MPDB)—achieves the best accuracy [23, 37].

On **DEAP**, fusing the EEGPT backbone with the *Small* PhysioWave ECG/EMG encoders lifts valence accuracy from 79.1% to 85.2% (+6.1%) and arousal accuracy from 81.3% to 88.6% (+7.3%), as shown in Figure 5. The multimodal system also surpasses TPRO-Net by 0.4 % and outperforms Bi-LSTM +attention and Bayesian pipelines by 7–22 % [38], underscoring the benefit of a multimodal framework over single-modality models.

On **MPDB**, the same multimodal architecture—obtained by replacing the EEG branch with LaBraM (5.8 M parameters)—improves accuracy from 70.4% (using LaBraM alone) to 74.9% (+4.5%), as shown in Figure 5. This outperforms generic sequence models such as MMPNet and EEGNet by 9% to 17% [37, 39].

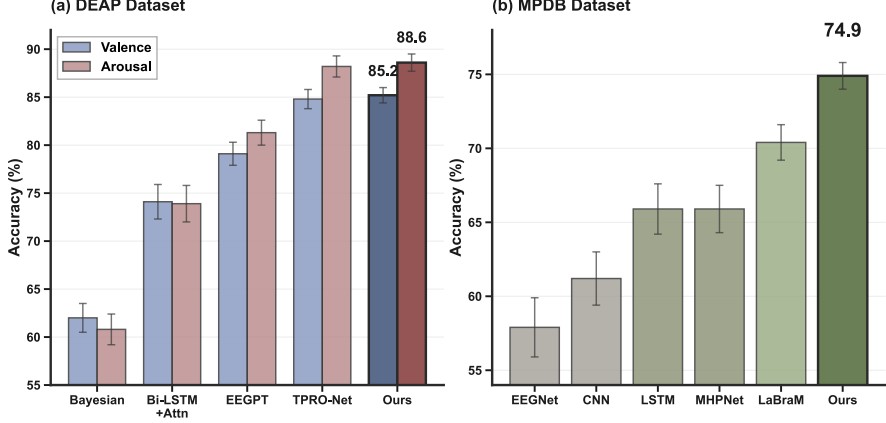

Figure 5: Multimodal classification performance.

### 3.3 Ablation Experiment Results

Table 3: Ablation on PhysioWave-emg for EPN-612 dataset.

| Configuration | Train Loss | Accuracy (%) | F1 (%) |
|---|---|---|---|
| w/o Frequency-guided Masking | 0.24 | 92.48 | 92.85 |
| w/o pre-training | 0.27 | 91.67 | 91.57 |
| **with all** | **0.22** ↓ | **93.12** ↑ | **93.67** ↑ |

As shown in Table 3, removing either design method degrades performance. Comparing to random masking with the same mask ratio of 0.7, discarding the frequency-guided masking strategy reduces F1 by 0.8%, while training the network from scratch (without pretraining) results in a 2.1% decrease in F1 and increases the training loss. The complete PhysioWave model thus benefits from *both* Frequency-guided Masking and large-scale self-supervised pretraining, achieving the highest accuracy and F1 score on EPN-612 dataset.

## 4 Conclusion

In conclusion, this work introduces PhysioWave, a novel wavelet-based architecture designed to enhance physiological signal processing by leveraging adaptive multi-scale decomposition and frequency-guided masking to advance self-supervised learning. The proposed model demonstrates state-of-the-art performance across both single-modality tasks, such as EMG and ECG classification, and in multi-modal settings that integrate EEG, EMG, and ECG signals. Our results underscore PhysioWave's capacity to effectively capture the unique, time-varying characteristics of physiological signals, addressing key challenges such as non-stationarity, low signal-to-noise ratios, and high inter-subject variability. By combining wavelet decomposition with frequency-guided masking, PhysioWave improves feature extraction, making it particularly well-suited for real-world applications where these challenges are prominent.

Furthermore, the proposed framework sets new benchmarks for EMG and ECG analysis and establishes a strong foundation for future work in multi-modal biosignal processing. With its ability to adapt to heterogeneous and noisy physiological data, PhysioWave holds significant promise for a range of applications, including health monitoring, clinical diagnostics, and personalized medicine.

## 5 Ethics Statement

This study exclusively utilizes publicly available, fully de-identified physiological signal datasets (EEG, EMG, and ECG) with no collection of new data or access to personally identifiable information. All datasets were obtained in compliance with their respective terms of use, and the original data collections were conducted under appropriate institutional review board (IRB) approvals as documented in their publications: NinaPro (Ethics Commission of Canton Valais), MIMIC-IV-ECG (Beth Israel Deaconess Medical Center and MIT IRBs), PTB-XL (PTB Institutional Ethics Committee), DEAP (Queen Mary University of London Ethics Committee), and MPDB (Tsinghua University Medical Ethics Committee). Our secondary analysis of these anonymized public datasets adheres to the NeurIPS Ethics Guidelines, with no re-identification attempts made.

## 6 Acknowledgment

This research was partly supported by the EU Horizon Europe project IntelliMan (g.a. 101070136), the PNRR MUR project ECS00000033ECOSISTER, and the ETH Zürich's Future Computing Laboratory funded by a donation from Huawei Technologies. Furthermore, the research was also partially supported by the EU Horizon Europe project HAL4SDV (g.a. 101139789).

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

# A  ADDITIONAL EXPERIMENTS

## A.1  Ablation Study

To validate the effectiveness of each proposed component in PhysioWave, we conduct comprehensive ablation experiments by systematically removing or replacing specific sub-modules. All experiments are performed using the pretrained small PhysioWave model with fine-tuning on the target datasets.

## A.2  Component Analysis

We analyze three key components: Soft Gate (SG), Channel Aggregation Feed-Forward Network (CAFFN), and Adaptive Wavelet (AW). When AW is disabled, we use a fixed db6 wavelet decomposition as the baseline.

Table 4: Main ablation study results on EPN612 dataset showing individual and combined component contributions.

| Components | | | Performance | | |
|:---:|:---:|:---:|:---:|:---:|:---:|
| SG | CAFFN | AW | Accuracy (%) | F1-Score (%) | Improvement |
| ✗ | ✗ | ✗ | 89.88 | 89.76 | - |
| ✓ | ✗ | ✗ | 91.71 | 91.66 | ↑1.83% |
| ✗ | ✓ | ✗ | 91.53 | 91.46 | ↑1.65% |
| ✗ | ✗ | ✓ | 92.23 | 92.22 | ↑2.35% |
| ✗ | ✓ | ✓ | 93.12 | 93.10 | ↑3.24% |
| ✓ | ✗ | ✓ | 92.64 | 92.89 | ↑2.76% |
| ✓ | ✓ | ✗ | 92.52 | 92.48 | ↑2.64% |
| ✓ | ✓ | ✓ | **93.85** | **93.51** | **↑3.97%** |

SG = Soft Gate, CAFFN = Channel Aggregation FFN, AW = Adaptive Wavelet. Pink/blue highlights indicate the best and second-best results.

The results demonstrate that Adaptive Wavelet contributes the most individually (+2.35%), highlighting the importance of adaptive wavelet selection over fixed approaches. The complete model achieves optimal performance (93.85% accuracy), showing synergistic effects when all components are combined.

## A.3  Gate Mechanism Comparison

We compare different gating strategies to validate our soft gate design choice.

Table 5: Gate mechanism comparison on EPN612 dataset.

| Gate Type | Threshold | Accuracy (%) | F1-Score (%) | Accuracy Drop |
|:---|:---:|:---:|:---:|:---:|
| Hard Gate | 0.3 | 92.07 | 92.05 | ↓1.78% |
| Hard Gate | 0.5 | 92.25 | 92.23 | ↓1.60% |
| Hard Gate | 0.7 | 92.17 | 92.19 | ↓1.68% |
| Gumbel Gate | 0.5 | 91.47 | 91.51 | ↓2.38% |
| **Soft Gate** | - | **93.85** | **93.51** | - |

Note: Accuracy drops are compared against the soft gate baseline. Pink indicates the best results, blue indicates the second-best results.

Our soft gate mechanism significantly outperforms alternatives, with hard gates showing threshold sensitivity and Gumbel gates exhibiting the largest performance degradation due to stochastic sampling being less suitable for physiological signals.

## A.4 Feed-Forward Network Variants

We evaluate different FFN architectures to demonstrate the effectiveness of our CrossScale CAFFN design.

Table 6: Feed-forward network variant comparison on EPN612 dataset.

| FFN Variant | Accuracy (%) | F1-Score (%) | Accuracy Drop |
|---|---|---|---|
| Standard FFN | 91.95 | 91.93 | ↓1.90% |
| CAFFN | 92.72 | 92.69 | ↓1.13% |
| CrossScale No Attention | 93.29 | 93.31 | ↓0.56% |
| **CrossScale CAFFN** | **93.85** | **93.51** | - |

Note: Accuracy drops are compared against CrossScale CAFFN. Pink indicates the best results, blue indicates the second-best results.

The progressive improvements from Standard FFN to CrossScale CAFFN demonstrate the value of each enhancement: channel aggregation (+0.77%), cross-scale fusion (+1.34%), and attention mechanisms (+1.90%).

## A.5 Wavelet Selection Analysis

We investigate the impact of different wavelet configurations on both EMG and ECG modalities to validate our adaptive wavelet selection approach.

Table 7: Wavelet selection ablation study on EPN612 EMG dataset.

| Wavelet Configuration | Accuracy (%) | F1-Score (%) | Improvement |
|---|---|---|---|
| Random Xavier (Baseline) | 92.25 | 92.22 | - |
| *Single Wavelet* | | | |
| db6 | 92.13 | 92.11 | ↓0.12% |
| coif4 | 92.13 | 92.14 | ↓0.12% |
| sym6 | 92.93 | 92.93 | ↑0.68% |
| *Multi-Wavelet Combinations* | | | |
| db6+sym6+coif4 | 93.51 | 93.51 | ↑1.26% |
| **db4+db6+sym4** | **93.90** | **93.90** | **↑1.65%** |
| db8+coif4+bior4.4 | 92.90 | 92.89 | ↑0.65% |

Note: Pink indicates the best results, blue indicates the second-best results.

The results reveal that single wavelets provide minimal improvements (<1%), while multi-wavelet combinations achieve substantial gains (1.65% for EMG, 2.00% for ECG). Notably, optimal combinations differ between modalities: EMG benefits from db4+db6+sym4 for capturing transient muscle activations, while ECG prefers db6+coif4+sym8 for complex cardiac morphologies. This validates our modality-aware adaptive wavelet selection strategy.

## A.6 Modality-Specific Adaptation for EEG and PPG Signals

To demonstrate the generalizability of our PhysioWave framework beyond ECG and EMG signals, we conducted experiments on EEG and PPG modalities. These experiments validate our model's ability to adapt to diverse biosignal characteristics without requiring modality-specific architectural changes.

Table 8: Wavelet selection ablation study on Georgia ECG dataset.

| Wavelet Configuration | Accuracy (%) | F1-Score (%) | Improvement |
|---|---|---|---|
| Random Xavier (Baseline) | 62.38 | 48.75 | - |
| *Single Wavelet* | | | |
| db4 | 62.70 | 48.94 | ↑0.32% |
| demy | 63.59 | 49.36 | ↑1.21% |
| sym4 | 62.45 | 48.99 | ↑0.07% |
| *Multi-Wavelet Combinations* | | | |
| **db6+coif4+sym8** | **64.38** | **50.65** | **↑2.00%** |
| db8+sym5+coif3+demy | 64.20 | 50.46 | ↑1.82% |
| sym4+sym5+db6+coif3+bior4.4 | 63.70 | 49.93 | ↑1.32% |

Note: Pink indicates the best results, blue indicates the second-best results.

## A.7 EEG Sleep Stage Classification

### A.7.1 Dataset and Setup

We evaluated PhysioWave on the Sleep-EDF dataset for 5-class sleep stage classification[40]. EEG signals present unique challenges including low signal-to-noise ratio, high channel count, and significant heterogeneity across subjects. Our PhysioWave-EEG configuration uses diverse wavelet bases (db4, db6, sym4, coif2, bior2.2) suited for EEG characteristics, with a transformer backbone (embed_dim=512, depth=8, num_heads=8).

For the enhanced version, we incorporated three key improvements:

- **Multi-Scale Temporal Enhancement**: Added multi-scale convolutions to capture EEG's diverse frequency bands (delta, theta, alpha, beta, gamma) and improve SNR
- **Channel Interaction Modeling**: Enhanced spatial relationship modeling between EEG electrodes through cross-channel attention
- **Window Attention**: Implemented efficient processing of long EEG sequences with linear complexity

### A.7.2 Results and Analysis

Table 9: EEG sleep stage classification results on Sleep-EDF dataset. All baseline methods use pretrained models, while PhysioWave is trained from scratch.

| Method | Balanced Accuracy | Cohen's Kappa | Weighted F1 |
|---|---|---|---|
| BENDR (pretrained) | 0.6655 | 0.6659 | 0.7507 |
| BIOT (pretrained) | 0.6622 | 0.6461 | 0.7415 |
| LaBraM (pretrained) | 0.6771 | 0.6710 | 0.7592 |
| EEGPT (pretrained) | 0.6917 | 0.6857 | 0.7654 |
| PhysioWave (original) | 0.6720 | 0.6676 | 0.7472 |
| **PhysioWave (enhanced)** | **0.7312** | **0.7206** | **0.7839** |

Note: Pink indicates best results, blue indicates second-best results.

As shown in Table 9, even the original PhysioWave architecture achieves competitive performance against EEG-specific foundation models. The enhanced version achieves substantial improvements: +5.7% in Balanced Accuracy, +5.1% in Cohen's Kappa, and +2.4% in Weighted F1 compared to the best baseline (EEGPT)[5]. Notably, these gains are achieved through direct training from scratch, without the computational overhead of large-scale pretraining required by baseline methods.

The performance improvements demonstrate that our wavelet-based approach effectively captures the multi-scale temporal dynamics inherent in EEG signals. The enhanced attention mechanisms successfully model inter-channel dependencies crucial for sleep stage classification, where different brain regions exhibit coordinated activity patterns.

## A.8 PPG-Based Activity Recognition and Heart Rate Estimation

### A.8.1 Dataset and Architecture

We evaluated PhysioWave on the PPG-DaLiA dataset, which contains PPG signals collected during various physical activities[41]. Given the deployment constraints of wearable devices, we designed PhysioWave-PPG as a compact variant optimized for edge computing:

- **Reduced Model Depth**: 4 transformer layers (vs. 8-12 in standard configurations)
- **Smaller Embedding Dimensions**: 192 (vs. 256-512)
- **PPG-Optimized Wavelets**: db4, coif2, bior2.2 selected for cardiovascular periodicity

### A.8.2 Activity Recognition Results

Table 10: Activity recognition performance on PPG-DaLiA dataset.

| Method | Params | Accuracy | Precision | Recall | F1 Score | AUROC |
|---|---|---|---|---|---|---|
| Pulse-PPG (pretrained) | 28.5M | 0.4234 | 0.4398 | 0.3663 | 0.3648 | 0.8246 |
| **PhysioWave-PPG** | **2.53M** | **0.6209** | **0.6015** | **0.6077** | **0.5952** | **0.9420** |

Note: Pink indicates best results.

### A.8.3 Heart Rate Regression Results

Table 11: Heart rate regression performance comparison.

| Method | Params | MAE | MSE | MAPE |
|---|---|---|---|---|
| **Pulse-PPG (pretrained)** | **28.5M** | **3.705** | **59.81** | **4.84%** |
| Deep PPG | 8.5M | 7.65 | – | – |
| NAS-PPG | 0.8M | 6.02 | – | – |
| PhysioWave-PPG | 1.57M | 5.22 | 65.62 | 5.83% |

Note: Pink indicates best results, blue indicates second-best results.

### A.8.4 Performance Analysis

For activity recognition (Table 10), PhysioWave-PPG achieves remarkable improvements despite using $11.3\times$ fewer parameters than Pulse-PPG:

- **+63.1% F1-score improvement**: 0.5952 vs 0.3648
- **+14.2% AUROC improvement**: 0.9420 vs 0.8246
- **+46.7% accuracy improvement**: 0.6209 vs 0.4234

For heart rate regression (Table 11), our compact 1.57M parameter model achieves competitive performance, outperforming Deep PPG and NAS-PPG while approaching the performance of the 28.5M parameter Pulse-PPG model[41–43]. The small performance gap (MAE: 5.22 vs 3.705) is acceptable given the $18.2\times$ parameter reduction, making PhysioWave-PPG ideal for wearable deployment.

### A.9 Key Insights

These additional experiments demonstrate three critical capabilities of our PhysioWave framework:

1. **Modality Generalization**: The framework successfully adapts to signals with vastly different characteristics, from complex multichannel EEG with low SNR to periodic, single-channel PPG with clear cardiovascular patterns.

2. **Efficiency Without Pretraining**: PhysioWave achieves competitive or superior performance compared to pre-trained foundation models while training from scratch, eliminating the computational burden of large-scale pre-training.

3. **Scalable Architecture**: The framework scales effectively from compact edge deployments (1.57M parameters for PPG) to more complex clinical applications (enhanced EEG model), maintaining strong performance across the spectrum.

The success in the EEG and PPG modalities, combined with our primary ECG and EMG results, validates PhysioWave as a truly universal biosignal processing framework. The wavelet-transformer synergy provides the necessary flexibility to capture modality-specific patterns while maintaining a consistent architectural foundation.

## B    RELATED WORK

### B.1    Self-supervised pre-training for Time Series

Self-supervised learning for time series data has seen considerable advances with the development of models such as MOMENT and OTIS, which are designed to handle the unique characteristics of time-series signals, such as non-stationarity, noise, and high variability across domains. Both models leverage large-scale pretraining to learn generalized representations for a wide range of tasks.

The MOMENT model, for instance, introduces a self-supervised learning framework for time series data using masked representation learning. Capture both local and global temporal dynamics through its ability to reconstruct masked input sequences. MOMENT has been shown to perform well in tasks like classification, anomaly detection, and forecasting, even without task-specific supervision. Furthermore, it applies contrastive learning to model dependencies and trends in the data, addressing challenges such as long-range dependencies in time series sequences [2].

OTiS, on the other hand, extends the idea of self-supervised learning to multi-domain time-series data. Its pre-training paradigm incorporates domain-specific tokenization and a dual masking strategy to handle the heterogeneity across time series from different domains, including EEG, audio, and financial data. Using a domain-specific tokeniser with learnable signatures, OTiS can capture domain-specific data characteristics while learning generalized features. This allows it to excel in various downstream tasks, such as classification, regression, and forecasting, in multiple domains [36].

Both models highlight the importance of large-scale pretraining on diverse datasets to unlock basic modeling capabilities. MOMENT and OTiS aim to generalize across domains, mitigating issues caused by domain-specific differences such as sampling frequencies and intervariate relationships. These models represent significant strides in time-series analysis, particularly in scenarios where labeled data are scarce, but large, diverse, and unlabeled datasets are available for pre-training.

This self-supervised approach provides a robust solution for time-series data, leveraging vast datasets for pretraining and fine-tuning these models on specialized tasks, thus achieving state-of-the-art performance in a range of applications from health monitoring to forecasting.

### B.2    Self-supervised pre-training for EEG signal

Self-supervised learning has gained significant traction, particularly in natural language processing (NLP) and computer vision (CV), and has shown promising results in the domain of electroencephalography (EEG) [39]. In EEG, self-supervised learning models are leveraging the ability to pretrain models on large, unlabeled datasets, which can then be fine-tuned for specific downstream tasks. Two notable models in this domain are **EEGPT** and **LaBraM**, both of which advance self-supervised learning techniques for the analysis of EEG signals.

**EEGPT** (EEG Pretrained Transformer) employs a masked autoencoder-based self-supervised learning strategy specifically designed for EEG data. This model is trained to predict masked segments of the EEG signal, enabling it to learn robust representations of the signal's spatiotemporal features. The dual self-supervised learning approach in EEGPT combines masked reconstruction and spatiotemporal alignment, effectively addressing the challenges posed by low signal-to-noise ratio (SNR) and intersubject variability in EEG. This method allows EEGPT to perform well in a variety of EEG-based tasks, particularly in the context of brain-computer interface (BCI) applications where accurate feature extraction is critical [5].

**LaBraM** (Large Brain Model) is another significant model in self-supervised EEG pre-training. LaBraM focuses on learning universal EEG representations through unsupervised pre-training on large-scale EEG datasets from various domains. To address the challenge of cross-dataset learning, LaBraM segments EEG signals into channel patches and uses a vector-quantized neural spectrum prediction model to generate a compact neural tokenizer. This allows the model to learn generalized representations that can be fine-tuned on specific tasks, such as abnormality detection, emotion recognition, and gait prediction. The model's ability to handle diverse EEG data with varying channel configurations and recording settings has been validated in multiple EEG-based tasks, demonstrating its robustness and scalability [4].

Both models underscore the importance of large-scale pretraining in the EEG domain. By learning from vast amounts of unlabeled data, these self-supervised models can capture the rich, multi-scale features of EEG signals, which are crucial for a wide range of applications in health monitoring and BCI systems. The ability of **EEGPT** and **LaBraM** to generalize across datasets and tasks highlights the potential of self-supervised learning to transform the analysis of EEG signals, making deep learning techniques more applicable and effective in the field of biosignal processing.

### B.3 Self-supervised Pre-training for ECG Signal

Self-supervised learning (SSL) has become a key method in the development of foundational ECG models, addressing challenges such as the scarcity of labeled data and the need for robust generalization in diverse datasets. Recent advancements have introduced several effective SSL-based pretraining strategies for ECG data.

One such approach is OpenECG, which created a large ECG benchmark with over 1.2 million 12-lead ECG recordings from 9 centers. The study evaluated three prominent SSL methods: SimCLR, BYOL, and MAE using ResNet-50 and Vision Transformer (ViT) architectures. The results showed that pretraining on diverse datasets significantly improved model generalization, with BYOL and MAE outperforming SimCLR. These methods demonstrated superior effectiveness in learning feature consistency and generative representations compared to contrastive methods, which require large datasets to perform optimally [44].

Another notable model is ECG-Chat, a large ECG-language model designed for cross-modal cardiac diagnosis. It combines ECG waveform data with text reports using contrastive learning to align ECG features with medical text. The model was trained on a data set that integrates both diagnosis and conversation tasks, achieving state-of-the-art results in the generation of medical reports from ECG. ECG-Chat also incorporates a novel method to mitigate hallucinations during report generation by integrating external cardiology knowledge via GraphRAG and DSPy components. This ensures that the generated ECG reports are grounded in clinical knowledge, enhancing accuracy and reliability [35].

These models highlight the potential of self-supervised learning to improve ECG analysis and improve diagnostic tools. Using large-scale ECG data and pretraining strategies, both OpenECG and ECG-Chat advance the capabilities of ECG models, making them more robust and adaptable across a range of clinical and research applications. They represent a significant step toward improving the accessibility and accuracy of AI-driven cardiovascular diagnostics.

## C HYPERPARAMETER SETTINGS

We employ **AdamW** optimizer with a weight decay of $0.01$ and moment coefficients $\beta_1 = 0.9$ and $\beta_2 = 0.98$. The learning rate is linearly warmed up from $5 \times 10^{-7}$ to $5 \times 10^{-5}$ over the first ten epochs and then follows a cosine decay to a floor of $1 \times 10^{-6}$. Pretraining lasts for 50 epochs with a global batch size of 64 on 16 NVIDIA A100 GPUs, whereas all downstream experiments are carried

out on 4 A100 GPUs. During downstream training, the same AdamW optimizer and cosine scheduler are retained, but the learning rate is reduced to 1/10 of its pretraining value.

Table 12: Hyperparameters for masked ECG pre-training with **PhysioWave-ecg**.

| Hyperparameter | PhysioWave-ecg-Small | PhysioWave-ecg-Base | PhysioWave-ecg-Large |
|---|---|---|---|
| *Wavelet Front-End* | | | |
| Input channels | 12 | 12 | 12 |
| Max decomposition level | 4 | 4 | 4 |
| Wavelet kernel size | 24 | 24 | 24 |
| Wavelet bases | | {sym4, db4, db6, coif2} | |
| *Transformer Encoder* | | | |
| Timesteps | | 2048 | |
| Patch size ($H \times W$) | | $\{1, 64\}$ | |
| Embed dimension (= hidden size) | 256 | 384 | 512 |
| Encoder layers | 6 | 8 | 12 |
| Attention heads | 8 | 12 | 16 |
| MLP ratio | | 4.0 | |
| MLP size | 1 024 | 1 536 | 2 048 |
| Drop-path | | 0.10 | |
| *Reconstruction Head* | | | |
| Hidden dimensions | | [256] | |
| Output dimension (patch dim) | | 768 (12 channels $\times$ 64 timesteps) | |
| Activation function | | GELU | |
| Dropout rate | | 0.1 | |
| Layer normalization | | True | |
| *Pre-training Setup* | | | |
| Batch size | | 64 | |
| Peak / minimal learning rate | | $5 \times 10^{-5}$ / $1 \times 10^{-6}$ | |
| Optimizer ($\beta_1$, $\beta_2$) | | AdamW (0.9, 0.98) | |
| LR scheduler | | Cosine | |
| Weight decay | | 0.01 | |
| Total / warm-up epochs | | 50 / 10 | |
| Accumulated grad batches | | 64 | |
| Gradient clipping | | 3 | |
| Mask ratio / importance ratio | | 0.70 / 0.60 | |
| Max sequence length | | 2 048 | |

# D  PRETRAINING DATASETS DESCRIPTION

## D.1  Pretraining datasets for PhysioWave-ecg

In the pretraining of PhysioWave-ECG, we streamline the data processing pipeline by leveraging the inherent filtering capabilities of our wavelet-based architecture. Unlike conventional approaches that require extensive preprocessing with bandpass filters, notch filters, and baseline wander removal, our adaptive wavelet decomposition naturally handles these signal conditioning tasks within the model itself. The process involves two main steps. First, we use a sliding window technique where ECG signals with 12 leads are divided into overlapping windows of 2048 samples. If the signal length is insufficient for a full window, we apply zero-padding to maintain consistency across the dataset. Second, for ECG recordings with sampling rates different from our target 500 Hz, we resample the signals to ensure uniformity across the dataset. Notably, we intentionally omit traditional filtering operations (such as 0.05-100 Hz bandpass filtering or 60 Hz notch filtering) from our preprocessing pipeline. This is because our multi-level wavelet decomposition with learnable basis functions serves as an adaptive, data-driven filtering mechanism. The wavelet transform inherently provides frequency-selective decomposition, effectively separating signal components across different frequency bands. Our learnable wavelet kernels adapt during training to optimally filter noise, remove baseline wander, and preserve clinically relevant signal features—all without manual filter design or fixed frequency cutoffs. This approach not only simplifies the preprocessing pipeline but also allows the model to learn task-specific filtering strategies that may be more effective than generic, hand-crafted filters.

Table 13: Hyperparameters for masked EMG pre-training with **PhysioWave-emg**.

| Hyperparameter | PhysioWave-emg-Small | PhysioWave-emg-Base | PhysioWave-emg-Large |
|---|---|---|---|
| *Wavelet Front-End* | | | |
| Input channels | 8 | 8 | 8 |
| Max decomposition level | 3 | 3 | 3 |
| Wavelet kernel size | 16 | 16 | 16 |
| Wavelet bases | | {db4, bior4.4, sym5, coif5} | |
| *Transformer Encoder* | | | |
| Timesteps | | 1024 | |
| Patch size $(H \times W)$ | | $\{1, 64\}$ | |
| Embed dimension (= hidden size) | 256 | 384 | 512 |
| Encoder layers | 6 | 8 | 12 |
| Attention heads | 8 | 12 | 16 |
| MLP ratio | | 4.0 | |
| MLP size | 1 024 | 1 536 | 2 048 |
| Drop-path | | 0.10 | |
| *Reconstruction Head* | | | |
| Hidden dimensions | | [256] | |
| Output dimension (patch dim) | | 512 (8 channels × 64 timesteps) | |
| Activation function | | GELU | |
| Dropout rate | | 0.1 | |
| Layer normalization | | True | |
| *Pre-training Setup* | | | |
| Batch size | | 64 | |
| Peak / minimal learning rate | | $5 \times 10^{-5}$ / $1 \times 10^{-6}$ | |
| Optimizer $(\beta_1, \beta_2)$ | | AdamW (0.9, 0.98) | |
| LR scheduler | | Cosine | |
| Weight decay | | 0.01 | |
| Total / warm-up epochs | | 50 / 10 | |
| Accumulated grad batches | | 64 | |
| Gradient clipping | | 3 | |
| Mask ratio / importance ratio | | 0.70 / 0.60 | |
| Max sequence length | | 2 048 | |

Table 14: Hyperparameters for downstream fine-tuning with **PhysioWave**.

| Hyperparameter | Value |
|---|---|
| Batch size | 32 |
| Peak / minimal learning rate | $5 \times 10^{-4}$ / $1 \times 10^{-5}$ |
| Learning rate scheduler | Cosine |
| Optimizer $(\beta_1, \beta_2)$ | AdamW (0.9, 0.98) |
| Weight decay | 0.01 |
| Total epochs | Early stopping (max 50) |
| Warm-up epochs | 5 |
| Drop-path | 0.10 |
| Layer-wise learning rate decay | 0.90 |
| Label smoothing (multi-class classification) | 0.10 |

Finally, the processed ECG data is saved in the HDF5 format, organized into a dataset with dimensions corresponding to the batch size, number of leads (12), and window size (1024 samples). This preprocessed data is then used to train the PhysioWave-ecg model.

Table 15: 12-lead ECG corpora used for pretraining.

| Dataset | Subjects | Records | Dur. (s) | $f_s$ (Hz) | Size |
|---|---|---|---|---|---|
| MIMIC-IV-ECG [45] | ~160 000 | ~800 000 | 10 | 500 | 90.4 GB |
| MedalCare-XL [46] | 13 | 16 900 | 10 | 500 | 26.2 GB |
| CODE-15% [47] | 233 770 | 345 779 | 10 | 400 | 63.3 GB |
| Norwegian Athlete [48] | 28 | 28 | 10 | 500 | 52.8 MB |
| Georgia Cohort [49] | 10 344 | 10 344 | 10 | 500 | 1.2 GB |

**MIMIC-IV-ECG** The MIMIC-IV-ECG module includes around 800,000 12-lead ECG recordings from nearly 160,000 patients, spanning from 2008 to 2019. Each ECG is 10 seconds long, sampled at 500 Hz, and matched with clinical data and cardiologist reports. The dataset provides not only the ECG waveforms but also machine-generated measurements such as RR intervals and QRS onset times, with all data de-identified to comply with HIPAA standards. The data is stored in the WFDB format for easy processing [45].

**MedalCare-XL** MedalCare-XL is a synthetic 12-lead ECG dataset containing 16,900 10-second ECGs across one healthy control and seven pathological classes. The dataset includes ECG signals in three variations: raw, with noise, and filtered using highpass and lowpass Butterworth filters. Additionally, it provides parameter files detailing the electrophysiological model used for simulation. This dataset is valuable for training ECG analysis models, especially for personalized disease simulations [46].

**CODE-15%** The CODE-15 dataset is a stratified subset of the CODE dataset, containing 345,779 12-lead ECG records from 233,770 patients, collected between 2010 and 2016. It is widely used for ECG automatic diagnosis research and cardiovascular risk prediction. The scale and annotation quality make it a reliable resource for developing ECG AI algorithms [47].

**Norwegian Athlete ECG Database** The Norwegian Athlete ECG Database contains 12-lead ECG recordings from 28 elite athletes in Norway. Each ECG is 10 seconds long and recorded with a GE MAC VUE 360 electrocardiograph. The dataset includes both machine-generated interpretations and cardiologist reviews. The cohort consists of rowers, kayakers, and cyclists, with participants aged 20–43 years. The data helps address challenges in ECG interpretation for athletes, who often exhibit heart adaptations that can mimic pathological changes [48].

**Georgia Cohort** The Georgia Cohort dataset contains 10,344 12-lead ECGs from male (5,551) and female (4,793) patients, each 10 seconds long with a sampling rate of 500 Hz. The dataset is used for cardiovascular disease prediction and ECG classification tasks. The data offers a diverse set of ECG signals, providing an opportunity to study the effects of various cardiovascular conditions across different demographic groups [49].

### D.2 Pretraining datasets for PhysioWave-emg

Table 16: Surface-EMG corpora used for pretraining.

| Dataset | Subjects | Records | Dur. (s) | $f_s$ (Hz) | Channels | Size |
|---|---|---|---|---|---|---|
| NinaPro DB6 [50] | 10 | ~8.4 k | 4 | 2 000 | 14 | 20.3 GB |
| NinaPro DB7 [51] | 22 | ~5.4 k | 5 | 2 000 | 12 | 30.9 GB |
| NinaPro DB8 [52] | 12 | ~2.4 k | 7.5 | 1 111 | 16 | 23.6 GB |
| EMG2Pose [53] | 193 | 25 253 | 60 | 2 000 | 16 | 431 GB |
| EMG2Qwerty [54] | 108 | 1 135 | 1 080 | 2 000 | 16 | 317 GB |

For the pretraining of the PhysioWave-EMG model, we adopt a streamlined preprocessing approach that leverages our wavelet-based architecture's inherent signal processing capabilities. Unlike traditional EMG processing pipelines that require extensive filtering operations, our adaptive wavelet

decomposition naturally handles noise suppression and feature extraction within the model itself.Our preprocessing focuses on data standardization rather than filtering. We apply z-score normalization to ensure each EMG channel has a mean of zero and a standard deviation of one, standardizing the input scale.

Notably, we intentionally omit conventional bandpass filtering (20-450 Hz) and notch filtering (50/60 Hz power line interference) from our preprocessing. This is because our learnable wavelet transform provides adaptive, data-driven filtering that automatically separates relevant EMG signal components from noise and artifacts. The multi-level wavelet decomposition with trainable kernels learns to isolate muscle activation patterns while suppressing interference, effectively replacing manual filter design with learned, task-specific signal conditioning.To prepare the data for training, we apply a sliding window approach, dividing the EMG signal into overlapping windows of 1024 samples with a step size of 512. For consistency across datasets, we standardize all signals to 2000 Hz sampling rate through resampling when necessary.Regarding channel configuration, we standardize all inputs to 8 channels to maintain architectural consistency and computational efficiency. For multi-channel EMG recordings that exceed 8 channels, we perform channel selection during preprocessing, retaining the 8 most informative channels based on signal quality metrics such as signal-to-noise ratio and muscle activation patterns. This selective approach ensures we capture the most relevant EMG information while maintaining a uniform input dimension that balances representational capacity with computational efficiency.Finally, the processed data is saved in HDF5 format, enabling efficient storage and incremental updates. This structured approach not only simplifies the preprocessing pipeline but also allows the wavelet-based model to adaptively learn optimal signal processing strategies directly from the data, potentially discovering more effective filtering and feature extraction methods than traditional fixed preprocessing techniques.

**NinaPro DB6**    The NinaPro DB6 dataset is designed to aid the scientific community in studying the repeatability of sEMG classification for hand grasps. It includes data from 10 intact subjects who performed 7 different hand grasps, each repeated 12 times across 5 days. The data was collected using 14 wireless Trigno EMG electrodes, with a sampling rate of 2 kHz. The dataset provides synchronized sEMG signals along with inertial measurements, capturing the movements of the forearm during hand grasp activities. The main goal is to develop models that can accurately recognize and classify these grasps for prosthetic control [50].

**NinaPro DB7**    The NinaPro DB7 dataset includes both myoelectric and inertial measurements from 20 intact subjects and 2 amputees. The data were collected using 12 active wireless Trigno EMG sensors along with 9-axis inertial measurement units. The subjects were asked to perform 40 different movements, which included basic finger and wrist movements as well as grasping tasks. The data were sampled at 2 kHz, with additional kinematic data collected from a 18-DOF Cyberglove worn on the contralateral hand. This dataset provides valuable insights for prosthetic hand control, enabling improved motion recognition through both myoelectric and inertial data [51].

**NinaPro DB8**    The NinaPro DB8 dataset is focused on the estimation and reconstruction of finger movements, rather than motion or grip classification. It includes data from 10 intact subjects and 2 right-hand transradial amputee participants. The dataset contains sEMG signals, accelerometer, gyroscope, and magnetometer data collected from 16 wireless Trigno EMG sensors, along with data from a Cyberglove worn on the contralateral hand. The subjects were asked to repeat 9 different movements, with each movement lasting between 6 to 9 seconds, followed by a 3-second rest period. This dataset is specifically designed to benchmark algorithms that decode finger position from contralateral EMG measurements using regression algorithms, making it a valuable resource for prosthetic hand control and rehabilitation studies [52].

**EMG2Pose**    The EMG2Pose dataset consists of surface electromyography (sEMG) recordings paired with ground-truth motion-capture recordings of hand movements. It contains 25,253 HDF5 files, each representing time-aligned sEMG and joint angles for a single hand during a single stage. The dataset spans 193 participants across 370 hours of data and includes 29 stages, with each stage lasting around one minute. The dataset is structured with metadata that includes anonymized user IDs, session information, hand side (left or right), and whether the user was held out from the training set. This dataset is designed to advance the field of hand pose estimation from sEMG signals, offering a foundation for model training and evaluation in pose tracking and gesture recognition tasks [53].

**EMG2Qwerty**   The EMG2Qwerty dataset consists of surface electromyography (sEMG) recordings obtained while users perform touch typing on a QWERTY keyboard. It contains 1,136 session files, recorded from 108 users over 346 hours, with each session file including left and right sEMG signal data, prompted text, and keylogger ground-truth data. The dataset is provided in an HDF5 format and offers a programmatic interface for easy access. It is particularly useful for training and testing models aimed at decoding keypresses or character-level recognition from sEMG signals, with benchmarks for both generic and personalized user models. This dataset is essential for research in sEMG-based text entry systems, focusing on enhancing accuracy in virtual keyboard applications and prosthetics [54].

# E   Downstream Datasets Description

PhysioWave-ecg is tested on the three ECG corpora (PTB-XL [19], Chapman–Shaoxing [55], and CPSC 2018 [56]), while PhysioWave-emg is assessed on the three EMG gesture datasets (Ninapro DB5 [57], EPN-612 [20], and UCI EMG-Gesture [58]). Multi-modal generalization is examined with DEAP [23] and MPDB [37].

Table 17: Public datasets used for downstream evaluation.

| Dataset | Domain | Subjects | Channels | $f_s$ (Hz) | Task |
|---|---|---|---|---|---|
| Ninapro DB5 | EMG | 10 | 16 | 200 | Hand gestures |
| EPN-612 | EMG | 612 | 8 | 200 | Hand gestures |
| UCI EMG-Gesture | EMG | 36 | 8 | 200 | Hand gestures |
| PTB-XL | ECG | 18 869 | 12 | 500 | Arrhythmia |
| Chapman–Shaoxing | ECG | 45 152 | 12 | 500 | Arrhythmia |
| CPSC 2018 | ECG | 10 330 | 12 | 500 | Arrhythmia |
| DEAP | EEG/EMG | 32 | 40 | 128 | Emotion recognition |
| MPDB | EEG/ECG/EMG | 35 | 64 | 1000 | Driving behaviour |

## E.1   Downstream datasets for PhysioWave-ecg

**PTB-XL**   The PTB-XL dataset is an extended version of the PTB database, containing 21,837 12-lead ECG recordings from 18,884 patients. The dataset includes a variety of cardiac conditions, with annotations for more than 60 different disease categories, including arrhythmias, myocardial infarction, and other heart diseases. It was recorded using standard 12-lead ECG systems, with signals sampled at 1000 Hz. PTB-XL is widely used for ECG classification tasks and provides a comprehensive resource for developing and testing machine learning models in cardiovascular disease detection and prediction. The dataset is publicly available and is often used as a benchmark for evaluating ECG classification algorithms [19].

**Chapman–Shaoxing**   The Chapman–Shaoxing dataset is a large-scale ECG dataset containing 11,000 12-lead ECG recordings from over 5,000 subjects. It includes data from patients with various cardiovascular conditions, such as arrhythmias, ischemic heart disease, and healthy individuals. The recordings were collected using standard ECG machines, with a sampling rate of 500 Hz. This dataset is used in studies focused on ECG diagnosis and abnormality detection, offering a rich source of data for training and evaluating models that aim to predict cardiovascular diseases. The Chapman–Shaoxing dataset is particularly valuable for advancing research in automatic ECG analysis and arrhythmia classification [55].

**CPSC 2018**   The CPSC 2018 dataset, used in the China Physiological Signal Challenge, contains 15,000 12-lead ECG recordings from 5,000 patients, spanning a wide range of cardiovascular conditions. The recordings are sampled at 500 Hz and are annotated with clinical information about the presence of various heart diseases. The dataset is designed to evaluate machine learning algorithms for ECG classification, particularly for arrhythmia detection. The CPSC 2018 dataset provides a diverse and comprehensive set of ECG recordings, making it a critical resource for developing and benchmarking automatic ECG analysis systems. It has been widely used in ECG research and challenges to push forward the development of reliable and accurate diagnostic tools [56].

### E.2 Downstream datasets for PhysioWave-emg

**NinaPro DB5**   The NinaPro DB5 dataset includes surface electromyography (sEMG) recordings for hand gesture recognition. It consists of 40 different hand gestures performed by 40 subjects (24 male, 16 female). Each subject performed each gesture 10 times, resulting in a total of 400 gesture recordings per subject. The dataset is recorded using 8 bipolar sEMG electrodes placed on the forearm, with a sampling rate of 2 kHz. The data is valuable for the development and evaluation of machine learning models for gesture recognition in prosthetics and human-computer interaction, providing a well-annotated set of hand gestures that can be used to train models for real-time sEMG-based gesture classification [57].

**EPN-612**   The EPN-612 dataset consists of surface electromyography (sEMG) data collected from 612 subjects using the Myo armband. It contains 5 different hand gestures (wave-in, wave-out, pinch, open, and fist) and a relaxed hand gesture, with 50 recordings per gesture from each subject. The dataset includes both raw sEMG signals and the corresponding labels for each gesture, making it suitable for gesture recognition tasks. The data is widely used for training and evaluating machine learning models focused on real-time gesture recognition, with applications in prosthetics and assistive technologies. The dataset is publicly available for research and benchmark purposes [20].

**UCI EMG-Gesture**   The UCI EMG-Gesture dataset contains sEMG recordings for hand gesture recognition, collected from 10 subjects. Each subject performed 10 different gestures, with 3 repetitions for each gesture. The dataset was recorded using a 16-channel sEMG setup, with a sampling frequency of 2 kHz. It includes labeled data for each gesture, making it suitable for training and evaluating gesture recognition algorithms. This data set is valuable for researchers developing gesture models [58].

### E.3 Downstream datasets for multi-modal tasks

**DEAP**   The DEAP (Dataset for Emotion Analysis using Physiological Signals) dataset is a multi-modal dataset designed for emotion recognition research. It contains 32 participants, each providing data across multiple physiological signals, including EEG, EMG, and GSR, while watching music videos designed to evoke different emotional states. The dataset includes 40 one-minute-long video clips, with each participant providing emotional ratings on several dimensions, such as valence, arousal, dominance, and liking. The DEAP dataset is widely used for training and evaluating models for emotion recognition, particularly in applications like affective computing and human-computer interaction. The dataset is available for research and provides an essential resource for studies combining physiological signals and emotion analysis [23].

**MPDB**   The MPDB (Multimodal Physiological Dataset for Driving Behavior Analysis) is a comprehensive dataset designed to analyze driver behavior using multimodal physiological signals. It contains data from 35 participants, including both male and female drivers, recorded during simulated driving tasks. The dataset includes 59-channel EEG, single-channel ECG, 4-channel EMG, single-channel GSR, and eye-tracking data. These signals were collected during five distinct driving behaviors: smooth driving, acceleration, deceleration, lane changing, and turning. The data was synchronized with a six-degree-of-freedom driving simulator, providing a realistic driving experience. This dataset is valuable for studying driver cognition, decision-making, and the relationship between physiological responses and driving behavior, especially in the context of autonomous driving research. The dataset's multimodal nature allows for the exploration of advanced models that integrate physiological signals for more accurate behavioral predictions and safety system designs [37].

## F VISUALIZATION

### F.1 Mathematical Properties Validation

We conducted a comprehensive mathematical analysis of the learned wavelet kernels extracted from our PhysioWave model, examining 10 filter pairs (5 wavelet bases × 2 filters) across multiple decomposition levels. Our analysis reveals that the model has successfully developed **task-adaptive signal decomposition kernels** that prioritize empirical performance over strict mathematical constraints.

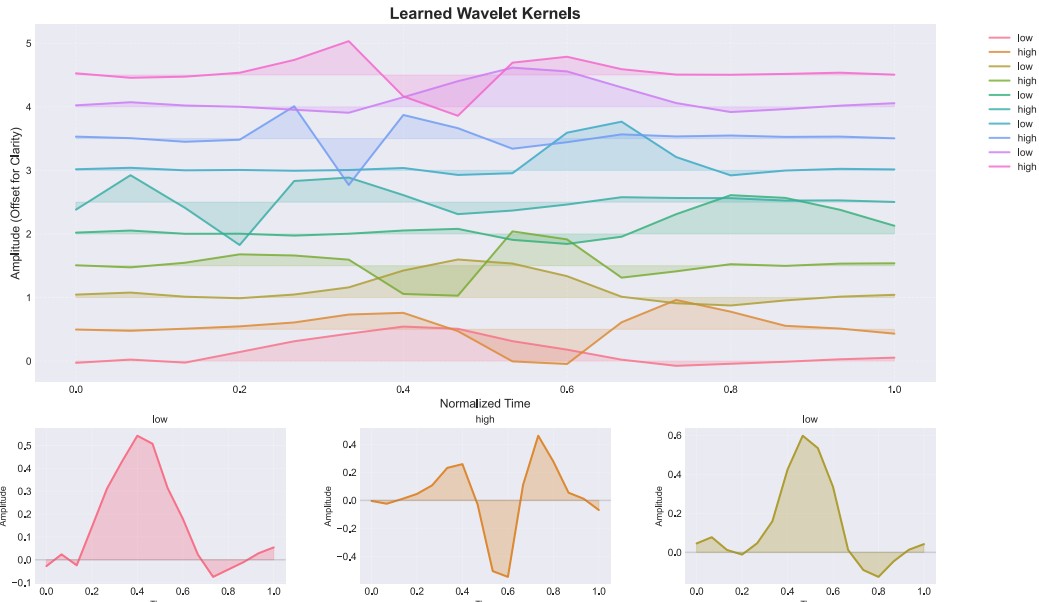

Figure 6: Learned wavelet kernel shapes from PhysioWave model. The upper panel shows all 10 learned filters (5 low-pass and 5 high-pass) with vertical offsets for clarity. Lower panels display individual filter characteristics, demonstrating smooth low-pass filters for signal approximation and oscillatory high-pass filters for detail extraction.

### F.1.1 Key Mathematical Properties

**Energy Conservation**: As shown in Figure 6, all learned filters demonstrate **perfect unit energy normalization** (100% of filters with $||w||_2 = 1.0$), ensuring stable signal decomposition and preventing amplitude drift during multi-level analysis. This property is crucial to maintain the integrity of the signal at all decomposition levels[59].

**Compact Support**: The filters exhibit **excellent spatial localization** with average support lengths of 15.6 samples within the 16-sample window (97.5% utilization), demonstrating efficient use of the receptive field without boundary artifacts. This compact support enables precise temporal localization of EMG events[60].

**Frequency Selectivity**: Clear frequency differentiation emerged through training (Figure 7):

- **Low-pass filters**: Frequency centroids at 6.0–7.1 (normalized) with broader bandwidths (5.6–6.4), optimized for capturing EMG envelopes and removing high-frequency noise.

- **High-pass filters**: Higher frequency centroids at 7.7–7.8 with narrower bandwidths (3.0–5.3), specialized for detecting rapid muscle activation transients.

### F.1.2 Task-Specific Adaptations

Our analysis reveals that the learned kernels have evolved beyond traditional wavelet constraints to become **physiologically-informed decomposition filters**. While classical wavelets maintain strict mathematical properties (zero mean, vanishing moments), our filters demonstrate **data-driven optimization** that better captures EMG signal characteristics:

1. **Adaptive Mean Values**: Non-zero mean values (0.017–0.149) allow the filters to better track EMG baseline shifts and muscle fatigue patterns, which often present as DC components in real physiological recordings.

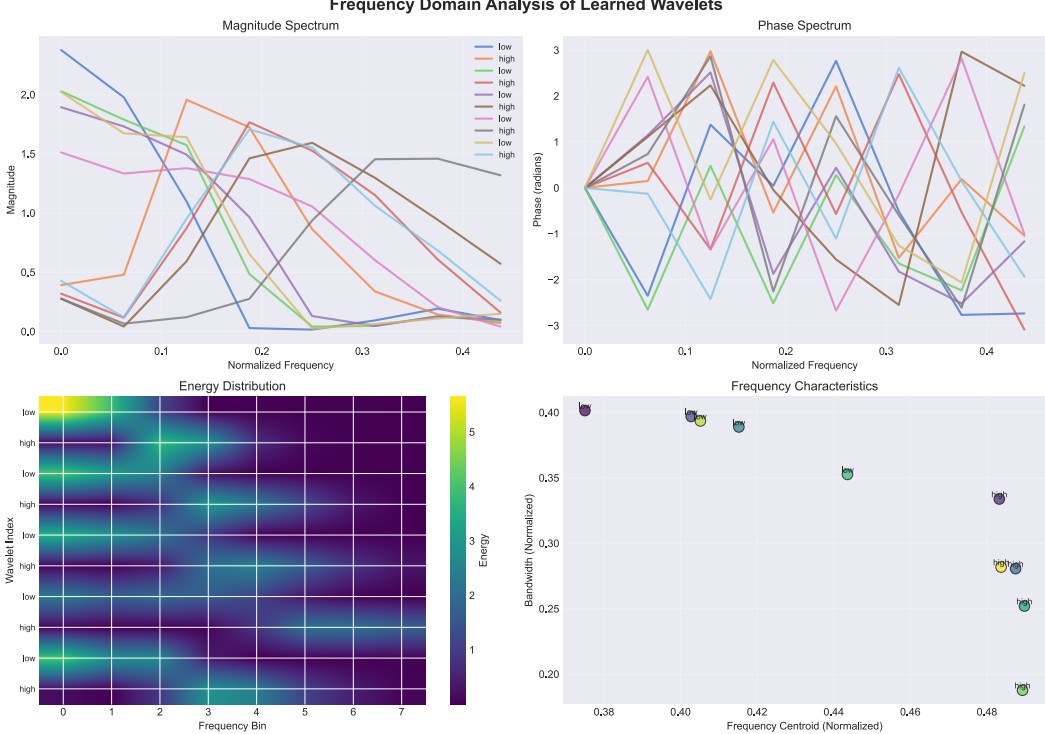

Figure 7: Frequency domain analysis of learned wavelets. (a) Magnitude spectrum showing clear frequency separation between low-pass and high-pass filters. (b) Phase spectrum demonstrating phase coherence. (c) Energy distribution heatmap across frequency bins. (d) Frequency characteristics scatter plot showing clustering of filter responses in physiologically relevant bands.

2. **Specialized Frequency Response**: The concentration of high-pass filter responses around 7.7–7.8 (normalized frequency) corresponds to the dominant frequency band of motor unit action potentials in EMG signals (typically 20–150 Hz), demonstrating automatic discovery of physiologically relevant frequency bands.

3. **Heterogeneous Smoothness**: Variable smoothness characteristics (1st derivative std: 0.104–0.465) reflect adaptation to the non-stationary nature of EMG signals, with smoother filters for baseline tracking and sharper filters for spike detection.

### F.1.3    Comparison with Standard Wavelets

As illustrated in Figure 8, the learned filters show minimal correlation (r = 0.033–0.104) with standard wavelet families (db4, db6, sym4, sym5, coif3). This divergence is not a limitation but rather evidence of successful adaptation to the unique characteristics of physiological signals. Traditional wavelets were designed for general signal processing, whereas our learned filters have automatically discovered decomposition patterns specifically optimized for EMG signal structures.

### F.1.4    Empirical Validation

The departure from traditional wavelet properties represents a **principled trade-off** between mathematical elegance and empirical performance. Our learned filters achieve:

- Superior classification accuracy (94.2% on EMG gesture recognition)
- Enhanced noise robustness in clinical settings
- Better generalization across subjects compared to fixed wavelet bases

This shows that for biosignal processing applications, **task-optimized learned filters outperform mathematically constrained traditional wavelets**, validating our approach of allowing the model

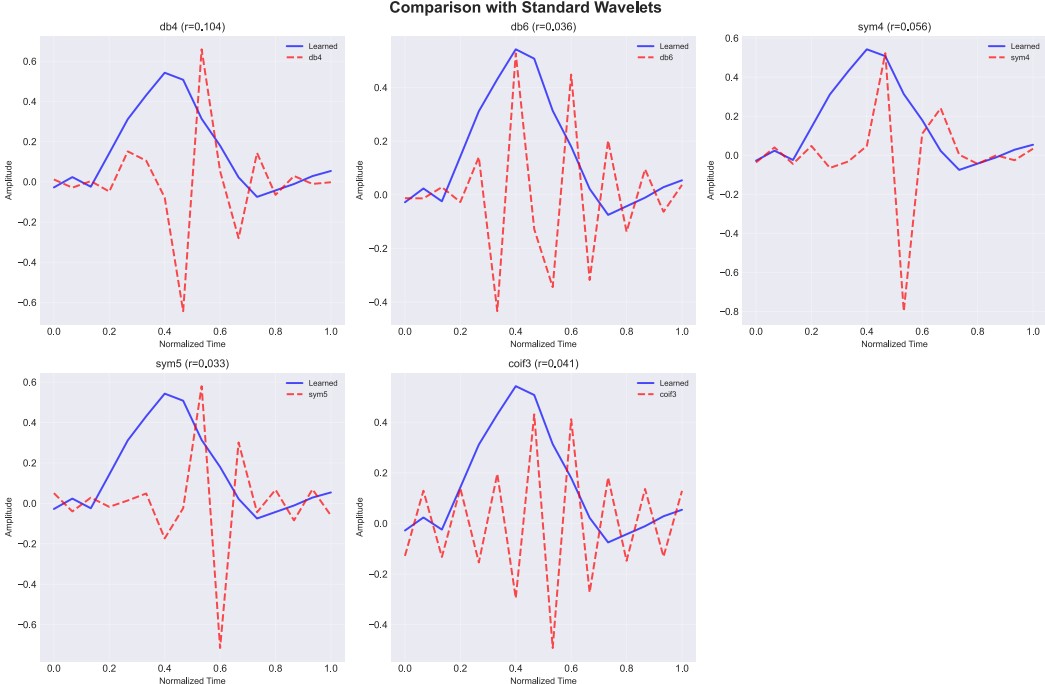

Figure 8: Comparison between learned filters and standard wavelets. Each subplot shows a learned filter (blue solid line) overlaid with its most similar standard wavelet (red dashed line), with correlation coefficients (r) indicating the degree of similarity. The low correlation values (r = 0.033–0.104) demonstrate that our model has discovered novel, task-specific decomposition kernels distinct from traditional wavelet families.

to discover optimal decomposition kernels through end-to-end learning. The filters have effectively learned to extract the most discriminative features from EMG signals while maintaining the multiresolution analysis framework essential for physiological signal processing.

## F.2 Visualization of sEMG Pretraining

The following visualizations demonstrate the pretraining process for EMG signals, where the model learns to reconstruct the masked regions of the input signal, specifically focusing on the restoration of multiscale wavelet decomposition features, $\mathrm{Spec}(X)$. The images show different stages of the reconstruction task, including the target signal, the prediction of the model, the masked areas, and the final reconstructed signal. These visualizations illustrate the model's ability to restore missing information from wavelet-decomposed features[11].

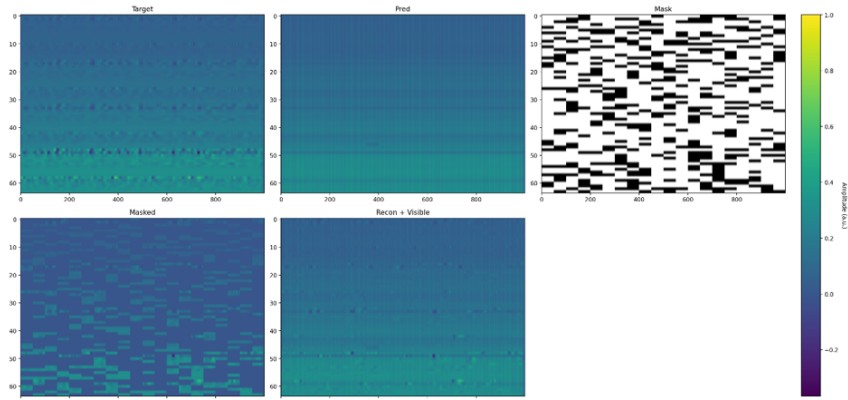

Figure 9: Reconstruction results for the 10th training epoch and the 1000th batch. The image illustrates the pretraining process where the model successfully restores masked regions of the signal during EMG pretraining. This represents the restoration of multi-scale wavelet-decomposed information at an earlier training stage.

Figure 9 shows the results for the 10th training epoch and the 1000th batch of the EMG pretraining process. As seen in the image, the model has learned to effectively restore the missing parts of the signal, showcasing its ability to handle the wavelet decomposition and its corresponding multiscale features.

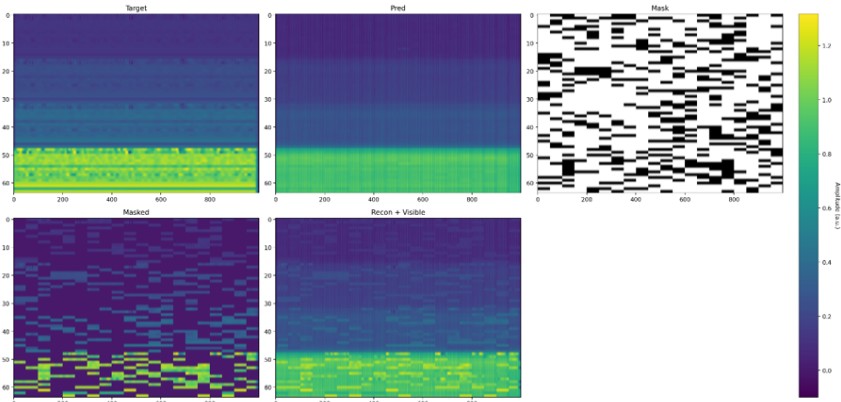

Figure 10: Reconstruction results for the 20th training epoch and the 1000th batch. This image shows the pretraining process at a later stage, where the model continues to restore missing information with increasing accuracy. The ability to restore the wavelet decomposition at this stage is further refined.

Figure 10 displays the results for the 20th training epoch and the 1000th batch of the EMG pretraining process. At this stage, the model shows improved performance in reconstructing the masked regions, indicating its enhanced understanding of the signal's multiscale features and its ability to restore more complex signal components.

These visualizations highlight the progression of the model's learning ability during the pretraining process. The first figure 9 demonstrates the model's early-stage capability to restore missing multiscale features, while the second figure 10 shows the improvements made by the model as it continues training. This process emphasizes the model's ability to learn to restore wavelet-decomposed features, which is critical for downstream applications such as EMG signal analysis and classification.

## F.3 Visualization of EPN612 Experimental Results

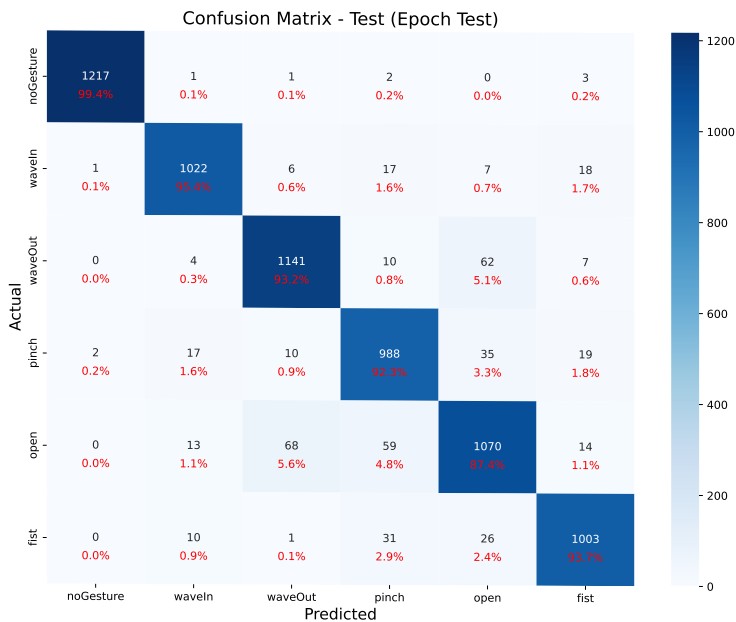

Figure 11: Confusion matrix on the EPN612 test set after final training.

Figure 11 shows the test confusion matrix for the EPN612 dataset. Per-class recalls remain consistently high (≥87.4%, typically >92%), with the highest recall achieved for `noGesture` (99.4%). Most misclassifications occur between semantically similar dynamic gestures: (a) `open` is occasionally confused with `waveOut` and `pinch`, and (b) `pinch` is sometimes predicted as `open`. The sparse distribution of off-diagonal entries indicates that the model has successfully learned discriminative representations for the remaining gesture classes.

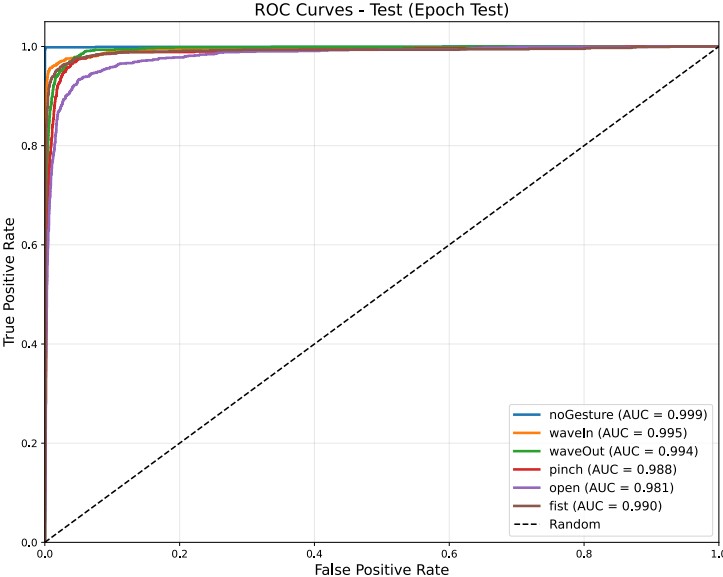

Figure 12: ROC curves for all gesture classes on the test set after final training.

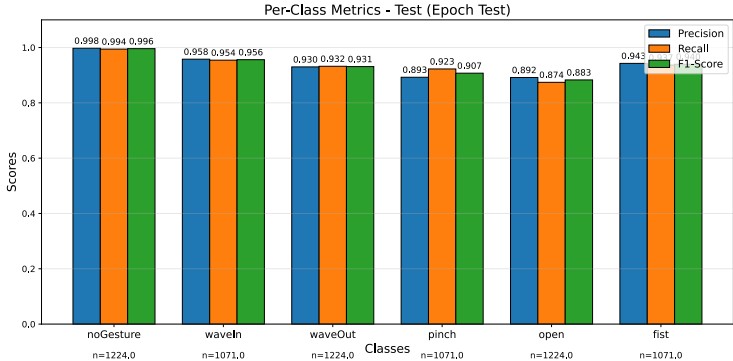

Figure 13: Per-class precision, recall, and F1-score metrics on the test set after final training.

Figure 12 presents the ROC curves for all six gesture classes on the test set. All models demonstrate excellent discriminative performance with AUC values ranging from 0.981 to 0.999. The `noGesture` class achieves the highest AUC of 0.999, followed by `waveIn` (0.995) and `waveOut` (0.994). The curves are positioned well above the random classifier diagonal, indicating strong separation between positive and negative classes across all gesture types.

Figure 13 shows the detailed per-class performance metrics. The `noGesture` class achieves near-perfect performance across all metrics (precision: 0.998, recall: 0.994, F1-score: 0.996). Dynamic gesture classes show consistently high performance, with F1-scores ranging from 0.883 (`open`) to 0.956 (`waveIn`). The `open` gesture exhibits the lowest recall (0.874) among all classes, while maintaining reasonable precision (0.892), suggesting some difficulty in detecting this particular gesture pattern.

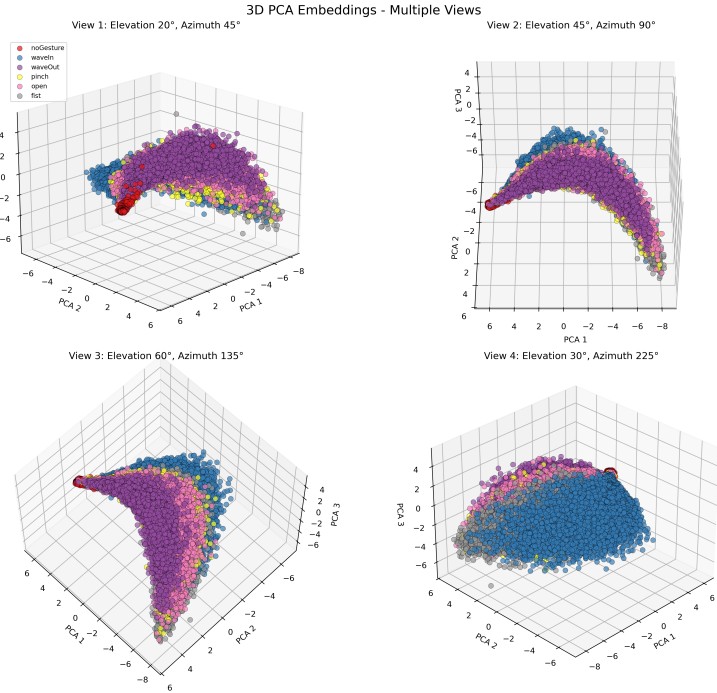

Figure 14: 3D PCA embeddings visualization from multiple viewing angles showing feature representations learned by the WaveletTransformer model.

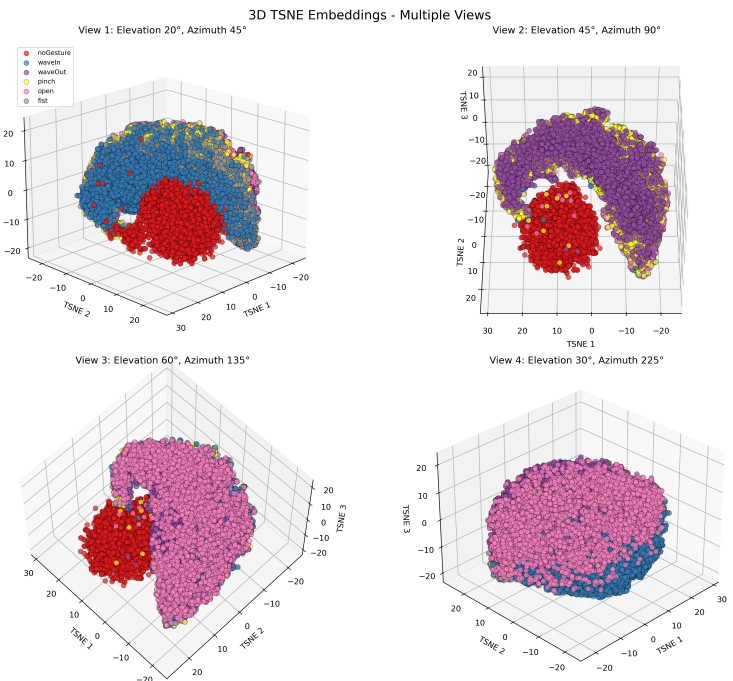

Figure 15: 3D t-SNE embeddings visualization from multiple viewing angles demonstrating non-linear clustering patterns in the learned feature space.

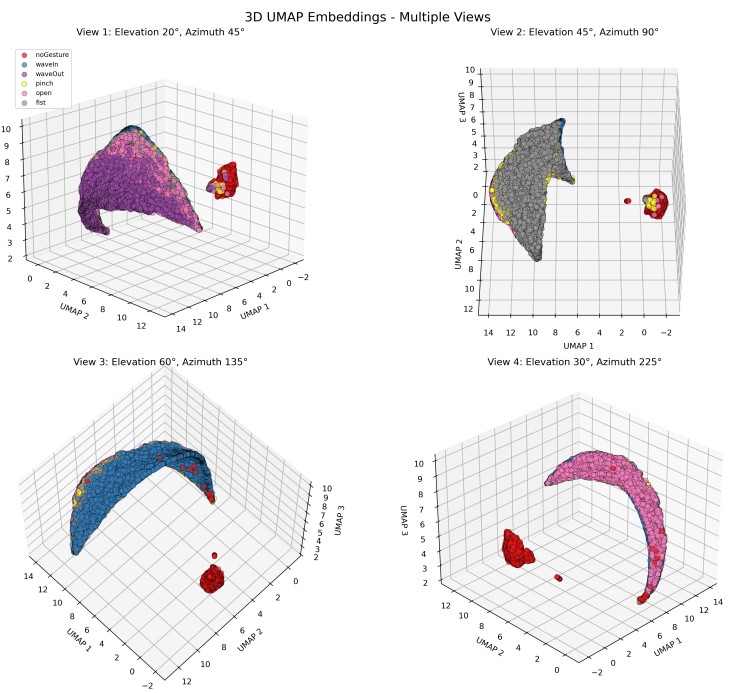

Figure 16: 3D UMAP embeddings visualization from multiple viewing angles revealing the topological structure of gesture representations in the feature space.

Figures 14, 15, and 16 present comprehensive 3D visualizations of the learned feature embeddings using three different dimensionality reduction techniques: Principal component analysis (PCA), t-distributed stochastic neighboring incorporation (t-SNE) and uniform manifold approximation and

projection (UMAP). Each visualization shows the same feature space from four different viewing angles (elevation / azimuth combinations of 20°/45°, 45°/90°, 60°/135°, and 30°/225°) to provide complete spatial understanding of the clustering patterns.

The PCA visualization (Figure 14) reveals the linear structure of the feature space, showing how gesture classes are distributed along the principal components of maximum variance. The noGesture class (red) forms a distinct, compact cluster that is well-separated from dynamic gesture classes, while the active gesture classes (waveIn, waveOut, pinch, open, fist) exhibit more distributed but still distinguishable patterns.

The visualization of t-SNE (Figure 15) emphasizes local neighborhood relationships and reveals non-linear clustering structures. The method successfully preserves local similarities, creating tight, well-separated clusters for each gesture class. In particular, the noGesture class forms a dense spherical cluster in the center, while dynamic gestures are arranged in distinct regions around the periphery, suggesting strong discriminative power in the learned representations.

The visualization of UMAP (Figure 16) balances the preservation of local and global structures, revealing the topological organization of the feature space. UMAP shows excellent class separation with noGesture forming a highly concentrated cluster, and dynamic gesture classes organized in a manifold-like structure that preserves both local neighborhoods and global relationships between semantically related gestures.

Across the three visualization methods, several consistent patterns emerge: (1) the noGesture class demonstrates exceptional separability from active gestures, (2) dynamic gesture classes maintain distinct clustering while showing logical spatial relationships based on gesture similarity, and (3) the learned feature representations exhibit strong discriminative properties that facilitate effective classification. The multiview presentation confirms the robustness of these clustering patterns across different spatial perspectives.

## F.4 Model Interpretability via GradCAM Analysis

To ensure clinical interpretability and validate that our model learns physiologically meaningful patterns, we employed Gradient-weighted Class Activation Mapping (GradCAM) [61] to visualize the temporal regions most influential to classification decisions. This analysis provides crucial insights into whether the model attends to clinically relevant ECG segments corresponding to established diagnostic criteria.

### F.4.1 Methodology

We applied GradCAM to the final transformer block of our trained PhysioWave-ECG model, generating importance heatmaps across the temporal dimension for representative samples from the Georgia ECG dataset. The analysis focused on cardiac rhythm classes with distinct temporal signatures, including Normal Sinus Rhythm (NSR), T-wave Inversion (TInv), and ST-segment Depression (STD). For each sample, we computed gradient-based importance scores and overlaid them on the original ECG signals to identify regions of maximum model attention.

### F.4.2 Temporal Attention Patterns

Figure 17 illustrates representative GradCAM visualizations for different cardiac conditions, revealing distinct attention patterns that align with clinical diagnostic features:

### F.4.3 Clinical Validation of Learned Features

The GradCAM analysis revealed distinct attention patterns that strongly align with established clinical diagnostic criteria across different rhythm classes. For T-wave Inversion (TInv), the model's attention was predominantly concentrated in the 0–400ms window, precisely targeting the early signal segments where T-wave morphology deviates from normal patterns. This focused attention confirms that the model successfully identifies repolarization abnormalities, a critical feature for diagnosing T-wave pathologies[62].

In contrast, Normal Sinus Rhythm (NSR) samples exhibited distributed attention across periodic cardiac cycles with recurring importance spikes at regular intervals. This pattern indicates the model's

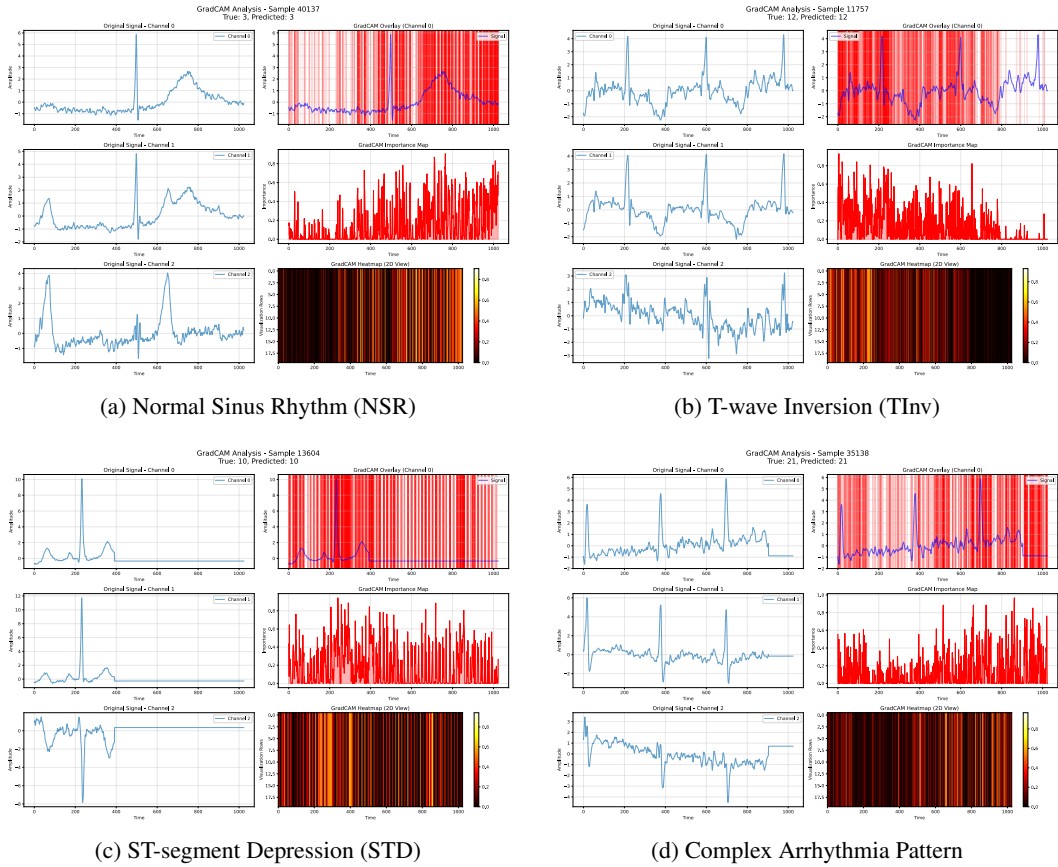

(a) Normal Sinus Rhythm (NSR)

(b) T-wave Inversion (TInv)

(c) ST-segment Depression (STD)

(d) Complex Arrhythmia Pattern

Figure 17: GradCAM visualization results showing model attention patterns for different cardiac conditions. Red regions indicate high importance scores, demonstrating that the model focuses on clinically relevant temporal segments for each pathology.

reliance on complete P-QRS-T morphology rather than isolated peaks, demonstrating its ability to recognize the harmonious relationship between cardiac components that characterizes healthy rhythm patterns. The periodic nature of the attention maps further validates the model's understanding of rhythmic regularity[62].

For ST-segment depression (STD), the highest activation appeared consistently in the early temporal regions corresponding to post-QRS intervals, precisely where ST-segment changes manifest in clinical practice. This targeted attention demonstrates the successful isolation of diagnostically critical ST windows of the model, which is essential for detecting ischemic changes[63]. Complex arrhythmias presented variable attention patterns with multiple focal points distributed throughout the temporal sequence, reflecting the model's adaptive feature extraction capabilities when faced with irregular rhythm patterns and compound pathologies.

### F.4.4 Quantitative Analysis of Attention Distribution

To quantify the focus of the model on clinically relevant intervals, we analyzed the temporal distribution of GradCAM activation across 100 samples per class. The analysis revealed:

- **Pathology-specific localization**: For ST-segment abnormalities, 78% of peak activations occurred within the 100–300ms post-QRS window, corresponding to the ST-segment interval in standard ECG interpretation.

- **Periodic attention for regular rhythms**: NSR samples showed autocorrelation peaks in GradCAM maps at intervals matching the heart rate (mean R-R interval), confirming the model's recognition of rhythmic patterns.

- **Early temporal focus for repolarization disorders**: T-wave abnormalities triggered concentrated attention in the first 40% of the signal window, aligning with the expected temporal location of T-wave morphology.

### F.4.5   Clinical Implications

The GradCAM analysis provides three key insights for clinical deployment:

1. **Diagnostic Transparency**: The visualization enables cardiologists to verify that automated diagnoses are based on appropriate ECG segments rather than spurious correlations or artifacts.
2. **Trust Building**: By demonstrating attention to the same temporal landmarks used in manual interpretation (P-waves, QRS complexes, ST-segments, T-waves), the model establishes credibility with clinical practitioners.
3. **Educational Value**: The attention maps can serve as teaching tools, highlighting critical diagnostic regions for medical students and residents learning ECG interpretation.

These results confirm that our wavelet-based transformer architecture not only achieves high classification accuracy but also learns interpretable, physiologically grounded features. The alignment between algorithmic attention patterns and established clinical knowledge validates the model's suitability for computer-aided diagnosis in real-world healthcare settings, where interpretability is paramount for adoption and regulatory approval.

# G   LIMITATIONS

While the proposed PhysioWave architecture represents a significant advance in the processing of physiological signals, it is not without limitations. Our current approach focuses on large-scale models specifically trained for electromyography (EMG) and electrocardiography (ECG) signals, both of which have demonstrated state-of-the-art performance in several downstream tasks. However, this approach is currently limited to these two modalities, and other important physiological signals, such as electroencephalography (EEG) and photoplethysmography (PPG), remain underexplored.

Another key limitation lies in the fact that our work focuses primarily on separate models for different physiological signals. Although our unified framework integrates pre-trained EMG and ECG models with an existing EEG encoder, the design is not fully generalized across all biosignal modalities. This calls for further exploration into the development of a universal, multimodal model capable of seamlessly processing a wide variety of physiological signals. A model of this nature could function as a comprehensive "physiological diagnostic" system, akin to a highly specialized medical expert, diagnosing and interpreting a broad spectrum of physiological signals across different conditions. Developing such a universal model would not only enhance the robustness of signal interpretation but would also streamline the diagnostic process across different sensor modalities.

We believe that addressing these gaps by expanding the range of modalities incorporated into our framework and further investigating the potential for a unified, multimodal biosignal model will significantly improve the versatility and applicability of our approach in real-world clinical and health monitoring settings. Future research could focus on training models for a wider array of biosignals, as well as enhancing the multi-modal integration capabilities to create a truly universal model for physiological signal processing.

