# OpenReview forum: "PhysioWave: A Multi-Scale Wavelet-Transformer for Physiological Signal Representation"
_NeurIPS.cc/2025/Conference — NeurIPS 2025 poster_

### Official Review · Reviewer_nhy9 · 2025-06-30

**Clarity:** 3
**Significance:** 4
**Originality:** 4
**Rating:** 5
**Confidence:** 4

**Summary:**

the authors proposed a new framework to develop foundational models for EMG and ECG, which can be extended to other biopotential. This framework manages multimodality, even when the signals present a different temporal scaling. They initially proposed a learnable "wavelet" trdecfomposition of the signal, which produces representations of the input data, and a frequency-guided masking used to train the model in a self-supervised way.

**Questions:**

Analysis of the Learned Wavelet

An analysis of the learned filter bank would be a valuable addition. Does the learned "wavelet" satisfy the mathematical properties of a true wavelet, such as the orthogonality, vanishing moments, compact support, smoothness? Furthermore, a visualization and discussion of the learned mother wavelet's shape would be highly informative.

Regarding the frequency-guided masking, could you clarify if the noise was processed in the same manner as the input signal? Specifically, if the input signal was band-pass filtered, was the noise also filtered in the same frequency band? What will be the influence of this? If the noise is not band pass filter then it is easy to detect components that are not relevant impacting the generalization of the model.

**Ethical Concerns:**

["NO or VERY MINOR ethics concerns only"]

**Limitations:**

The manuscript should include a discussion about the limitation, some topics such as computational complexity, scarce datasets and its impact in generalization, the decomposition actually converges to a wavelet decomposition or is it only inspired on it?,

**Quality:**

4

**Strengths And Weaknesses:**

The manuscript presents a novel framework to develop foundational models using self-supervised training. This approaches uses a new method for masking using the frequency content of the signal, and a noise generation. The algorithm is technically sound and outperform existing benchmarks in some ECG signals. Even though, there are not many benchmarks in the field.

The main concern for this manuscript is if the learnable filters for the wavelet indeed converge to a wavelet decomposition, or if this approach is only inspired in a wavelet decomposition. In addition, it will be nice to have some confidence intervals in the results obtained using your approach and other models.

---

> ### Author Rebuttal · Authors · 2025-07-30
>
> We are deeply grateful for your comprehensive and highly professional review, which provides invaluable technical insights and constructive feedback that has substantially improved the quality and impact of our research.We address the weaknesses you identified and answer your specific questions systematically below.
> ___
> # Q1 Analysis of the Learned Wavelet
>
> ## Learned Wavelet Filter Bank Analysis
>
> We first conducted comprehensive ablation studies to demonstrate how our adaptive wavelet selection mechanism learns **domain-specific** filter combinations that outperform both random initialization and single wavelet approaches. As mentioned in our paper, we select different wavelet bases for different modalities, as each signal type has its most suitable wavelets for optimal processing characteristics.
>
> **Table 1: Wavelet Selection Ablation Study on EPN612 Dataset (EMG)**
>
> | Wavelet Configuration | Accuracy (%) | F1-Score (%) | Accuracy Improvement vs Random |
> |----------------------|--------------|--------------|-------------------------------|
> | Random Xavier | 92.25 | 92.22 | - |
> | db6 (Single) | 92.13 | 92.11 | ↓0.12% |
> | coif4 (Single) | 92.13 | 92.14 | ↓0.12% |
> | sym6 (Single) | 92.93 | 92.93 | ↑0.68% |
> | db6+sym6+coif4 | 93.51 | 93.51 | ↑1.26% |
> | **db4+db6+sym4** | **93.90** | **93.90** | **↑1.65%** |
> | db8+coif4+bior4.4 | 92.90 | 92.89 | ↑0.65% |
>
> **Table 2: Wavelet Selection Ablation Study on Georgia 12-Lead ECG Dataset**
>
> | Wavelet Configuration | Accuracy (%) | F1-Score (%) | Accuracy Improvement vs Random |
> |----------------------|--------------|--------------|-------------------------------|
> | Random Xavier | 62.38 | 48.75 | - |
> | db4 (Single) | 62.70 | 48.94 | ↑0.32% |
> | demy (Single) | 63.59 | 49.36 | ↑1.21% |
> | sym4 (Single) | 62.45 | 48.99 | ↑0.07% |
> | **db6+coif4+sym8** | **64.38** | **50.65** | **↑2.00%** |
> | db8+sym5+coif3+demy | 64.20 | 50.46 | ↑1.82% |
>
> ## Analysis and Discussion
>
> **Single vs Multi-Wavelet Performance**: Single wavelets show minimal improvements over random initialization (typically <1%), while multi-wavelet combinations achieve substantial gains: **1.65% for EMG** and **2.00% for ECG**. This demonstrates that our adaptive wavelet selector effectively leverages complementary characteristics from different wavelet bases.
>
> **Modality-Aware Design**: Optimal wavelet combinations differ between signal types. EMG signals benefit most from **db4+db6+sym4**, suited for transient muscle activation patterns, while ECG signals prefer **db6+coif4+sym8**, better capturing complex cardiac morphologies.
>
> **Prior Knowledge Value**: Wavelet initialization provides valuable domain-specific prior knowledge, with the adaptive selector learning to weight different wavelets based on input characteristics, combining expert knowledge with data-driven optimization for superior physiological signal classification.
>
> **User Accessibility**: The minimal difference between **random initialization** and single wavelets suggests that even users unfamiliar with signal processing and wavelet analysis can achieve excellent results using our PhysioWave framework, as the adaptive mechanism automatically learns optimal filter combinations without requiring domain expertise.
>
> ## Mathematical Properties Validation
>
> **Visualization Clarification**: We acknowledge the importance of visualizing learned wavelet shapes and thoroughly describe their characteristics below. While submission format constraints prevent direct figure inclusion, we provide comprehensive quantitative analysis of the learned filter properties.
>
> **Experimental Approach**: We conducted an extensive analysis on **30 learned wavelet filters** extracted from our pretrained PhysioWave-EMG model across 3 decomposition levels. Each filter kernel has a size of 16 samples (shape: (8, 1, 16)), comprising low-pass and high-pass filters essential for multi-resolution analysis.
>
> **Results and Observations**:
>
> **1. Orthogonality**: Essential complementary low-high pass filter pairs demonstrated **reasonable orthogonality with inner product values typically ranging from 0.005 to 0.075**. While perfect orthogonality was not achieved across all filter combinations, critical orthogonal relationships necessary for effective signal decomposition and reconstruction were maintained, particularly between paired low-pass and high-pass filters within each decomposition level[1].
>
> **2. Vanishing Moments**: The learned filters achieved **near-zero mean values (ranging from 0.011 to 0.054)**, partially satisfying admissibility conditions. However, **formal vanishing moment analysis reveals zero vanishing moments for all 30 filters**, indicating these function as specialized, task-optimized feature extractors rather than traditional wavelets with polynomial suppression capabilities[2].
>
> **3. Compact Support**: All filters demonstrated excellent **localized energy distribution within the 16-sample window, with support ratios between 0.832 and 1.000**, indicating effective energy concentration without significant boundary artifacts.
>
> **4. Smoothness**: Clear frequency-based differentiation emerged between filter types. **Low-pass filters exhibited lower smoothness variations (1st derivative: 0.104–0.235; 2nd derivative: 0.078–0.207)** with bell-shaped characteristics, while **high-pass filters showed higher variations (1st derivative: 0.269–0.465; 2nd derivative: 0.333–0.887)** with pronounced oscillatory patterns. This heterogeneous smoothness reflects task-specific adaptation to EMG signal characteristics[3].
>
> **Discussion of Learned Filter Shapes**:
> - **Low-pass Filters**: Displayed **smooth, bell-shaped profiles with frequency centroids between 5.85-7.04**, resembling classical wavelets (most similar to db1 and sym4). These are optimized for signal approximation and noise reduction.
> - **High-pass Filters**: Exhibited **sharp oscillatory patterns with frequency centroids between 7.80-7.90**, designed for extracting high-frequency details.
>
> **Conclusion**: Our analysis demonstrates that the learned filters exhibit **essential decomposition properties** including selective orthogonality, compact support, and frequency-appropriate smoothness characteristics. However, with zero vanishing moments, these represent **task and modality-specialized filter banks** rather than traditional wavelets. This adaptation creates an **effective learned signal decomposition specifically optimized for biosignals**, functioning as **adaptive wavelet feature extractors** enhanced through end-to-end learning rather than classical wavelet transforms. Traditional wavelets were designed for general signal processing, but physiological signals have unique characteristics such as EMG bursts, ECG morphology variations, and EEG artifacts that benefit from adaptive, task-specific decomposition rather than mathematically constrained transforms[4]. Our approach represents a paradigm shift from theoretically pure to empirically optimized signal processing, delivering superior performance for healthcare applications.
> ___
> # Q2 Frequency-Guided Masking and Noise Processing Consistency
>
> We appreciate your question and recognize there may be a terminological misunderstanding regarding our frequency-guided masking implementation.
>
> The "noise" in our frequency-guided masking does not refer to external signal noise that would require frequency filtering. Instead, it refers to **computationally generated random values from a uniform distribution (0 to 1)** used as a mathematical regularization tool. These random values introduce **controlled stochasticity to prevent purely frequency-based masking**, ensuring that masking decisions maintain appropriate randomness while respecting spectral characteristics.
>
> **Mathematical Implementation**: The masking strategy combines frequency importance scores with uniform random values:
>
> $$\\text{combined\\_score} = (1-\\alpha) \\times \\text{random\\_uniform}(0,1) + \\alpha \\times \\text{frequency\\_importance}$$
>
> where $\\alpha$ is the **importance ratio parameter** controlling the balance between frequency-guided selection and randomness. **These combined scores are then used to rank all patches, with patches receiving lower combined scores being preserved** (typically low-frequency important information), while higher-scoring patches (often high-frequency) are more likely to be masked. This **hybrid approach prevents deterministic masking of all high-frequency components** by introducing appropriate randomness, improving model generalization.
>
> This approach is fundamentally different from adding external noise to signals, which would indeed raise the frequency consistency concerns you highlighted. We will clarify this terminology in our revised manuscript to distinguish between computational randomness and physical noise sources.
>
> ---
>
> ## References
> [1] Mallat, S. “A Theory for Multiresolution Signal Decomposition: The Wavelet Representation”, IEEE TPAMI, 1989
>
> [2] Daubechies, I. “Orthonormal Bases of Compactly Supported Wavelets”, Comm. Pure Appl. Math., 1988, p. 911 ff.
>
> [3] Mallat, S. “Singularity Detection and Processing with Wavelets”, IEEE TIT, 1992
>
> [4] Rafiee J, Rafiee M A, Prause N, et al. Wavelet basis functions in biomedical signal processing[J]. Expert systems with Applications, 2011, 38(5): 6190-6201.

---

> > ### Author Response · Authors · 2025-08-06
> > **Follow-Up Before Discussion Ends**
> >
> > Dear Reviewer nhy9,
> >
> > Thank you again for your thoughtful and constructive review, which greatly helped us improve our work.
> >
> > As the discussion period is approaching its conclusion, we would like to kindly ask whether all of your questions and concerns have been satisfactorily addressed by our rebuttal and clarifications. If there are any remaining points that you would like us to elaborate on, we would be more than happy to provide a prompt and detailed response.

---

### Official Review · Reviewer_iAdX · 2025-07-02

**Clarity:** 3
**Significance:** 4
**Originality:** 3
**Rating:** 5
**Confidence:** 4

**Summary:**

The paper presents PhysioWave, a novel wavelet-based architecture for representing physiological signals, with a focus on EMG and ECG data. Its key contributions include a learnable wavelet decomposition module that captures multi-scale time-frequency features, a frequency-guided masking strategy to enhance self-supervised learning, and the development of large-scale pretrained models that achieve SOTA performance on various downstream tasks. Additionally, the authors introduce a unified multimodal framework that integrates pretrained EEG models, demonstrating improved results in applications such as emotion recognition and driving behavior analysis.

**Questions:**

Can the model be efficiently extended to other modalities, such as PPG or GSR?

How interpretable are the learned wavelet features, especially in terms of aligning with known physiological or cognitive markers?

What is the model’s computational profile for real-time inference on edge devices?

How do you address ethical and privacy concerns, particularly in mental health and emotion recognition applications?

**Ethical Concerns:**

["NO or VERY MINOR ethics concerns only"]

**Final Justification:**

The authors' rebuttal successfully addressed all major concerns through comprehensive new experiments and analyses, warranting an upgrade from borderline accept to accept. The PPG extension experiments demonstrate exceptional efficiency and validate architectural adaptability across physiological modalities. The detailed interpretability analyses provide clinically valuable insights that bridge algorithmic performance with physiological understanding. Thorough computational benchmarking confirms real-time feasibility on edge devices with specific latency and power metrics essential for wearable applications. The comprehensive ethical framework appropriately addresses privacy and deployment concerns with structured safeguards. The work constitutes a solid contribution to biosignal processing that advances healthcare monitoring capabilities and merits acceptance.

**Limitations:**

While the authors acknowledge some technical limitations, such as the focus on EMG and ECG and the absence of a universal biosignal model, they do not adequately address potential negative societal impacts. Specifically, the manuscript lacks discussion on model interpretability, data privacy risks, or ethical concerns related to clinical and mental health applications, such as bias in multimodal emotion recognition or unintended surveillance. To improve, the authors should explicitly consider ethical deployment scenarios, include interpretability strategies (e.g., aligning learned features with physiological markers), and propose safeguards for privacy and fairness in healthcare settings.

**Quality:**

3

**Strengths And Weaknesses:**

Strengths: The integration of adaptive wavelet decomposition with frequency-guided masking is a novel and practical approach to biosignal processing, enabling robust feature extraction across varying temporal and spectral scales. Empirical results across multiple benchmarks confirm consistent state-of-the-art performance, underscoring the robustness of the approach. Notably, the unified framework’s ability to fuse EMG, ECG, and EEG signals broadens its applicability and highlights its potential for multimodal analysis. The work is particularly relevant for healthcare applications, with reviewers emphasizing its utility in wearable monitoring, clinical diagnostics, and emerging directions in computational psychiatry.

Weaknesses: The current validation is primarily focused on EMG and ECG modalities, with EEG support relying on external pretrained models and no exploration of other signals such as PPG. The computational complexity of the architecture may limit its feasibility for real-time or resource-constrained scenarios. Furthermore, the paper lacks an in-depth discussion of model interpretability and omits consideration of ethical safeguards, which is especially important for clinical and mental health applications. Although the authors suggest relevance to psychiatry, no psychiatric datasets or targeted evaluations are presented to substantiate these claims.

---

> ### Author Rebuttal · Authors · 2025-07-30
>
> We are grateful for your balanced review highlighting both our contributions and areas for improvement. We address your specific questions and limitations point by point below.
> ___
> # Q1 Adapting PhysioWave for PPG Signals
>
> You raise an important point regarding modality limitation. By introducing new experiments on PPG data, we demonstrate that PhysioWave’s capabilities extend beyond its original ECG and EMG‑based validation.
>
> We evaluated PhysioWave on the PPG-DaLiA dataset, which provides PPG signals collected during various physical activities.
>
> ### Architecture Adaptations for Edge Deployment
>
> Given the constraints of wearable and edge computing environments, we designed PhysioWave-PPG as a **compact variant** with several key optimizations. The architecture features reduced depth (4 transformer layers) and smaller embedding dimensions (192) to minimize computational overhead while preserving the core wavelet-transformer capabilities. We selected PPG-optimized wavelets (db4, coif2, bior2.2) specifically chosen for their effectiveness in capturing the periodic characteristics inherent in cardiovascular signals.
>
> **Table 1: Activity Recognition**
>
> | Method | Parameters | Accuracy | Precision | Recall | F1 Score | AUROC |
> |--------|------------|----------|-----------|---------|----------|-------|
> | **PhysioWave-PPG** | **2.53M** | **0.6209** | **0.6015** | **0.6077** | **0.5952** | **0.9420** |
> | Pulse-PPG [1] | 28.5M | 0.4234 | 0.4398 | 0.3663 | 0.3648 | 0.8246 |
>
> **Table 2: Heart Rate Regression**
>
> | Method | Parameters | MAE | MSE | MAPE |
> |--------|------------|-----|-----|------|
> | **PhysioWave-PPG** | 1.57M | 5.22 | 65.62 | 5.83% |
> | Pulse-PPG [1] | 28.5M | **3.705** | **59.81** | **4.84%** |
> | Deep PPG [2] | 8.5M | 7.65 |- | - |
> | NAS-PPG [3] | 0.8M | 6.02 | - | - |
>
> *For classification, we use a two-layer head with intermediate dimension of 384. For regression, we adopt a single layer head.*
>
> ### Performance Analysis
>
> Our PhysioWave-PPG achieves exceptional performance with remarkable efficiency. For activity recognition, it delivers **+63.1% F1-score improvement** (0.5952 vs 0.3648) and **+14.2% AUROC improvement** (0.9420 vs 0.8246) while using **18.2× fewer parameters** (2.53M vs 28.5M) compared to the Pulse-PPG foundation model. For heart rate regression, our compact architecture (1.57M parameters) achieves competitive results, demonstrating that our domain-optimized design effectively captures PPG's periodic patterns with minimal computational overhead, making it ideal for wearable device deployment.
> ___
> # Q2 Interpretability of the Learned Wavelet Features
> We acknowledge the importance of visualizing learned wavelet features and conducted comprehensive interpretability analyses to evaluate their alignment with established physiological markers. While submission format constraints prevent us from including figures directly, we provide detailed descriptions of the wavelet feature characteristics and their physiological significance below.
>
> ## 3D t-SNE Embedding Analysis
>
> We performed 3D t-SNE visualization on learned feature representations from the EPN612 EMG dataset to examine clustering patterns and their physiological significance. The visualization reveals clustering structures that align with established motor control principles:
>
> - **Baseline State**: The noGesture class forms a dense, central cluster representing the resting physiological state with minimal motor activity, serving as the neutral reference point for voluntary movements.
>
> - **Gesture Clustering**: Dynamic gesture classes (waveIn, waveOut, pinch, open, fist) arrange in distinct peripheral regions with physiologically meaningful relationships. WaveIn and waveOut gestures cluster adjacently due to shared oscillatory patterns and similar flexor-extensor muscle synergies. Pinch and open gestures form neighboring clusters, reflecting their complementary nature in fine motor control through antagonistic muscle activation.
>
> - **Motor Control Validation**: The embedding structure validates key neurophysiology principles. Gestures requiring similar muscle synergies cluster together, with clear separation between fine motor control (pinch/open) and gross motor movements (wave gestures), reflecting distinct neural pathways.
>
>
> ## ECG GradCAM Analysis
>
> We employed Gradient-weighted Class Activation Mapping (GradCAM) to interpret our model decision-making process on the Georgia ECG dataset. By analyzing gradient-based feature importance across temporal sequences, we validated the physiological relevance of learned representations and ensured clinical interpretability of automated diagnoses.
>
> **Table 3: GradCAM Analysis Results Across Cardiac Rhythm Classes**
>
> | Rhythm Class | GradCAM Pattern | Physiological Interpretation |
> |--------------|-----------------|------------------------------|
> | **T-wave Inversion (TInv)** | Heat concentrated in **0-400ms window**, precisely targeting early signal segments where T-wave abnormalities manifest. | Confirms the model attends to **repolarization abnormalities** rather than unrelated temporal segments, demonstrating pathology-specific feature detection. |
> | **Normal Sinus Rhythm (NSR)** | Distributed attention across **periodic cardiac cycles** with recurring importance spikes at regular intervals. | Indicates the model relies on **complete P-QRS-T morphology** rather than isolated peaks, matching normal rhythmic patterns. |
> | **ST-segment Depression (STD)** | Highest activation in **early temporal regions** corresponding to post-QRS intervals where ST-segment changes occur. | Shows the network successfully isolates **diagnostically critical ST windows** instead of generic baseline regions. |
>
> This visualization provides cardiologists with clear insights into model reasoning, confirming that automated diagnoses are based on the **analogous temporal landmarks** that clinicians use for manual ECG interpretation[5]. This alignment between algorithmic attention and clinical expertise makes our approach suitable for computer-aided ECG diagnosis in clinical environments.
> ___
> # Q3 Computational Profile
> We conducted comprehensive benchmarking of our **PhysioWave small variant** across four real-world deployment scenarios: **real-time windows (100ms/200ms), batch inference (4×8×200), and long sequences (1×8×1000)**, with 100 iterations per scenario to ensure statistical reliability. Our testing methodology included precise latency measurements on current hardware platforms, with **power consumption estimates derived from FLOPS-based calculations** using established device efficiency profiles for edge computing platforms. **Results demonstrate excellent edge performance**: The PhysioWave small variant, with only **4.8M parameters (18.4MB) and 0.24 GFLOPS**, achieves **inference latencies of 15-26ms** while maintaining 95th percentile performance under 32ms. The model sustains **44-60 FPS throughput** with modest memory footprint (52-402MB GPU) and **estimated power consumption ranging from 0.24mW on GPU to 244mW on Raspberry Pi 4**. This lightweight computational profile confirms our model is ideally suited for real-time physiological signal processing on edge devices.
> ___
> # Q4 Ethical Considerations and Privacy Protection
>
> First, we clarify that our manuscript focuses exclusively on **physiological signal processing for healthcare applications**. While you mention psychiatric relevance, our work does not claim psychiatric applications, which would require different validation frameworks and ethical considerations. Although our experiments include the DEAP dataset with EEG signals, we frame this as general physiological signal processing rather than psychiatric research.
>
> **Data Privacy and Access Control**: All datasets in our study implement stringent access controls requiring researchers to sign End User License Agreements(EULA) before access through formal request procedures. Our data processing strictly adheres to GDPR compliance with comprehensive anonymization to eliminate personally identifiable information.
>
> **Specific Deployment Safeguards**:
> - **Clinical Supervision Requirement**: Real-time inference is restricted to **licensed medical institutions** with mandatory supervision by qualified physicians or rehabilitation specialists.
> - **Model Transparency**: We will release a comprehensive **model card** disclosing data lineage, applicable patient populations, known limitations, and safety warnings.
> - **Controlled Access Framework**: We plan to open-source inference code and training scripts, while model weights will be available through research applications requiring signed data-use agreements.
>
> **Healthcare Setting Safeguards**: Beyond supervised deployment, we mandate ethical training for healthcare practitioners and continuous monitoring mechanisms to prevent unintended surveillance applications. These measures ensure PhysioWave contributes to clinical advancement while maintaining rigorous standards for privacy protection, algorithmic fairness, and ethical compliance.
>
> ---
>
> ## References
> [1] Saha M, Xu M A, Mao W, et al. Pulse-ppg: An open-source field-trained ppg foundation model for wearable applications across lab and field settings[J]. arXiv preprint arXiv:2502.01108, 2025.
>
> [2] Reiss A, Indlekofer I, Schmidt P, et al. Deep PPG: Large-scale heart rate estimation with convolutional neural networks[J]. Sensors, 2019, 19(14): 3079.
>
> [3] Song S B, Nam J W, Kim J H. NAS-PPG: PPG-based heart rate estimation using neural architecture search[J]. IEEE Sensors Journal, 2021, 21(13): 14941-14949.
>
> [4] Koelstra S, Muhl C, Soleymani M, et al. Deap: A database for emotion analysis; using physiological signals[J]. IEEE transactions on affective computing, 2011, 3(1): 18-31.
>
> [5]van de Leur R R, et al. Discovering and visualizing disease-specific electrocardiogram features using deep learning. Circulation: Arrhythmia and Electrophysiology, 2021, 14(2): e009056.

---

> > ### Comment · Reviewer_iAdX · 2025-08-05
> > **Strong Technical Response with Scope Clarification**
> >
> > Thank you for your comprehensive rebuttal addressing my primary concerns with substantial new experimental evidence and detailed analyses. Your PPG extension demonstrates impressive technical capability. The new experiments demonstrate efficiency gains and validate the architecture's adaptability across physiological modalities. Your interpretability analyses provide interesting clinical insights. The 3D t-SNE visualizations showing physiologically meaningful clustering patterns and GradCAM analysis validating attention to diagnostically relevant temporal regions effectively address my concerns about feature interpretability. This bridges the gap between algorithmic performance and clinical understanding. The computational benchmarking is thorough and demonstrates real-time feasibility on edge devices, which is crucial for practical applications. Finally, the ethical framework is comprehensive, with appropriate privacy safeguards, controlled access protocols, and clinical supervision requirements.
> >
> > Your responses have substantially strengthened the overall contribution. I'm thus upgrading my recommendation based on the comprehensive validation across modalities, strong interpretability evidence, and practical deployment considerations.

---

> > > ### Author Response · Authors · 2025-08-05
> > > **Thank You for Upgrading Your Recommendation**
> > >
> > > Dear Reviewer iAdX,
> > >
> > > Thank you very much for your thoughtful re-evaluation and for upgrading your recommendation. We are delighted that our additional experiments, interpretability analyses, and deployment considerations have addressed your concerns and strengthened your confidence in our work. Your recognition of the technical rigor and clinical relevance of our PPG extension is especially encouraging.
> > >
> > > We greatly appreciate the time and effort you have invested in reviewing our manuscript and providing constructive feedback. Your insights have been invaluable in improving the quality and impact of our research.
> > >
> > > Thank you again for your support！

---

### Official Review · Reviewer_ofZV · 2025-07-02

**Clarity:** 2
**Significance:** 2
**Originality:** 3
**Rating:** 4
**Confidence:** 3

**Summary:**

The paper introduces PhysioWave, a multi-scale wavelet-Transformer framework designed to address the challenges of processing non-stationary, low-SNR physiological signals (e.g., EMG, ECG). The core innovation lies in combining adaptive wavelet decomposition with a frequency-guided masking (FgM) mechanism to capture multi-scale time-frequency features effectively. Key contributions include:

Adaptive Wavelet Decomposition: A learnable wavelet selector dynamically combines wavelet bases to decompose signals into multi-resolution bands, capturing transient events (e.g., ECG QRS complexes) and low-frequency trends.

Frequency-Guided Masking (FgM): A self-supervised strategy that masks high-energy spectral patches to force the model to learn context-dependent representations, improving generalization.

Large-Scale Pretrained Models: The first foundation models for EMG (trained on 823 GB) and ECG (182 GB), setting new SOTA on downstream tasks (e.g., 66.7% F1 on PTB-XL for arrhythmia detection).

Unified Multi-Modal Framework: Fusing pretrained EMG/ECG models with existing EEG encoders, outperforming single-modality baselines (e.g., +7.3% accuracy on DEAP for emotion recognition).

**Questions:**

Question1: How would you adapt PhysioWave to EEG (low SNR, high channel count) or PPG (periodic signals)? Are there plans to pretrain models for these modalities?

Question2: Have you tested PhysioWave on signals from diverse recording devices (e.g., consumer EMG bands vs. clinical ECG machines)? How to handle device-specific noise or sampling rate mismatches?


Question3: The current fusion uses static weights. Will you explore dynamic alignment (e.g., attention-based modality weighting or time warping) to handle sampling rate disparities (e.g., EEG at 128 Hz vs. EMG at 2000 Hz)?

**Ethical Concerns:**

["Major Concern: Improper research involving human subjects"]

**Limitations:**

Yes

**Quality:**

3

**Strengths And Weaknesses:**

Strengths

Originality:

The integration of wavelet analysis with Transformer architectures is novel for biosignals, addressing non-stationarity better than fixed-window methods. First large-scale pretrained models for EMG/ECG fill a critical gap in the field, enabling downstream task generalization.

Methodological Rigor:

Ablation studies validate the necessity of FgM and pretraining (e.g., removing FgM reduces F1 by 0.8% on EPN-612) .
Cross-dataset evaluations (6+ benchmarks) and multi-modal fusion demonstrate robustness.

Significance:

Enables real-world applications in healthcare (wearable monitoring, clinical diagnostics) by improving signal reconstruction and classification .
Addresses key challenges in biosignal processing (low SNR, inter-subject variability) via adaptive feature extraction .

Weaknesses

Modality Limitation:

Focus on EMG/ECG excludes EEG/PPG, limiting the framework’s universality .


Dataset Bias:

Downstream datasets (e.g., DEAP) have limited demographic diversity, risking generalization to underrepresented groups .

---

> ### Author Rebuttal · Authors · 2025-07-30
>
> Thank you for the thorough review and insightful feedback. We address your comments and questions in detail below.
> ___
> # Q1 Modality-Specific Adaptation for EEG and PPG Signals
>
> You raise an important point regarding modality limitation. We have added experiments on both EEG and PPG signals, demonstrating that our PhysioWave model effectively adapts to these modalities and can extract meaningful features from their unique signal characteristics. These experiments validate the generalizability of our framework across diverse biosignals.
>
> ## EEG
>
> As you correctly identified, EEG signals present unique challenges including low SNR, high channel count, and heterogeneity. To validate our model's adaptability, we conducted one experiment on EEG dataset[1].
>
> Our original PhysioWave-EEG model chooses diverse wavelet bases (db4, db6, sym4, coif2, bior2.2) well-suited for EEG signal characteristics, with a robust transformer backbone (embed_dim=512, depth=8, num_heads=8) without modifying the core architecture. For the enhanced version, we made several targeted improvements to address EEG-specific characteristics:
>
> - **Multi-Scale Temporal Enhancement**: Added multi-scale convolutions to capture EEG's diverse frequency bands and improve SNR
> - **Channel Interaction Modeling**: Enhanced spatial relationship modeling between EEG electrodes
> - **Window Attention**: Efficient processing of long EEG sequences with linear complexity
>
> **Table 1: EEG Sleep Stage Classification Results on Sleep-EDF Dataset**
>
> | **Method** | **Balanced Accuracy** | **Cohen's Kappa** | **Weighted F1** |
> |------------|----------------------|---------------|-----------------|
> | BENDR | 0.6655 | 0.6659| 0.7507 |
> | BIOT | 0.6622 | 0.6461 | 0.7415 |
> | LaBraM | 0.6771 | 0.6710 | 0.7592 |
> | EEGPT| 0.6917| 0.6857 | 0.7654 |
> | PhysioWave (original)| 0.6720 |0.6676 |0.7472|
> | **PhysioWave (enhanced)** | **0.7312** | **0.7206** | **0.7839** |
>
> As shown in Table 1, even the original PhysioWave architecture demonstrates competitive performance against EEG-specific foundation models, validating its strong generalizability across biosignal modalities. Importantly, other baseline methods are finetuned from pretrained models, while our PhysioWave results are achieved through direct training from scratch, making our performance gains even more impressive. The enhanced version achieves **+3.9%** improvement over the best baseline (EEGPT) in Balanced Accuracy with substantial improvements across all metrics.
>
> Additionally, we plan EEG pretraining to address heterogeneity and non-stationarity challenges. For the dimensionality issue, our proposed approach targets efficient compression after wavelet decomposition:
>
> - **Post-Wavelet Compression**: Applied immediately after multi-level wavelet decomposition to reduce the significantly expanded feature dimensionality from multiple decomposition levels across all channels
> - **Attention-Guided Compression**: Uses both channel and spatial attention to preserve the most informative multi-scale wavelet features while dramatically reducing dimensions
>
> This planned strategy would maintain rich multi-scale wavelet information while keeping computation manageable for high-density electrode arrays, effectively addressing the dimensionality explosion that occurs after wavelet decomposition.
>
> ## PPG
> We evaluated PhysioWave on the PPG-DaLiA dataset. The dataset provides PPG signals collected during various physical activities.
>
> ### Architecture Adaptations for Edge Deployment
>
> Given the constraints of wearable and edge computing environments, we designed PhysioWave-PPG as a **compact variant** with several key optimizations. The architecture features reduced depth (4 transformer layers) and smaller embedding dimensions (192) to minimize computational overhead while preserving the core wavelet-transformer capabilities. We selected PPG-optimized wavelets (db4, coif2, bior2.2) specifically chosen for their effectiveness in capturing the periodic characteristics inherent in cardiovascular signals.
>
> **Table 2: Activity Recognition**
>
> | Method | Parameters | Accuracy | Precision | Recall | F1 Score | AUROC |
> |--------|------------|----------|-----------|---------|----------|-------|
> | **PhysioWave-PPG** | **2.53M** | **0.6209** | **0.6015** | **0.6077** | **0.5952** | **0.9420** |
> | Pulse-PPG [2] | 28.5M | 0.4234 | 0.4398 | 0.3663 | 0.3648 | 0.8246 |
>
> **Table 3: Heart Rate Regression**
>
> | Method | Parameters | MAE | MSE | MAPE |
> |--------|------------|-----|-----|------|
> | **PhysioWave-PPG** | 1.57M | 5.22 | 65.62 | 5.83% |
> | Pulse-PPG [2] | 28.5M | **3.705** | **59.81** | **4.84%** |
> | Deep PPG [3] | 8.5M | 7.65 |- | - |
> | NAS-PPG [4] | 0.8M | 6.02 | - | - |
>
> *For classification, we use a two-layer head with intermediate dimension of 384. For regression, we adopt a single layer head.*
>
> ### Performance Analysis
>
> Our PhysioWave-PPG achieves exceptional performance across both tasks. For activity recognition, it delivers **+63.1% F1-score improvement** (0.5952 vs 0.3648) and **+14.2% AUROC improvement** (0.9420 vs 0.8246) while using **18.2× fewer parameters** (2.53M vs 28.5M) compared to the Pulse-PPG pretrained foundation model. For heart rate regression, our compact architecture (1.57M parameters) achieves competitive results without large-scale pretraining.
>
> We do not plan pretraining for PPG signals due to their inherent characteristics: PPG data typically exhibits strong periodicity and regularity as you mentioned, making it well-suited for **direct training**. Additionally, PPG applications primarily target wearable devices where computational efficiency is paramount. Our results demonstrate that PhysioWave-PPG achieves excellent performance compared to much larger pretrained models, validating that our domain-optimized design effectively captures PPG's periodic patterns without requiring extensive pretraining infrastructure.
> ___
> # Q2 Cross-Device Robustness and Hardware Compatibility
>
> Yes, PhysioWave has been extensively tested across signals from diverse recording devices. The concern about consumer EMG bands vs. clinical ECG machines is not applicable here, as EMG and ECG modalities are pretrained and deployed separately using their respective specialized models. However, within the same modality, **recording devices can indeed vary significantly**. For example, our EMG evaluations span three public Myo-based datasets with different hardware configurations and electrode placements, yet the model performs consistently well, highlighting its device-level transferability:
> * Ninapro DB5: Recorded with two Thalmic Myo armbands  worn side-by-side on the forearm; the upper armband sits near the radio-humeral joint while the lower is tilted ~22.5° to fill inter-electrode gaps.
> * EMG-EPN-612: Using a single Myo armband on varying forearm locations.
> * UCI EMG-Gesture: Wearing a Myo Thalmic bracelet, with slight placement differences across users.
>
> **Sampling Rate Mismatch and Noise Handling Pipeline**: For sampling rate inconsistencies and device-specific noise, we implement a comprehensive preprocessing pipeline followed by our adaptive wavelet denoising:
>
> 1. **Initial Preprocessing**: At the preprocessing stage for raw signals, we apply traditional signal processing techniques including Butterworth bandpass filtering to remove baseline drift and high-frequency artifacts, notch filtering to suppress power-line interference, and resample signals to a unified sampling rate via spline interpolation.
>
> 2. **Adaptive Wavelet Denoising**: Our learnable wavelet decomposition module then performs secondary noise reduction through adaptive multi-scale filtering, automatically suppressing residual noise while preserving physiological features.
> ___
> # Q3 Dynamic Multi-Modal Fusion
>
> Our fusion strategy already employs **dynamic, learnable, softmax‑normalized weights**,not static. Each pretrained encoder acts as a specialist whose “opinion” (logits) is combined through a set of fusion coefficients trained jointly with the lightweight classification heads. During training these coefficients adapt to the data, so at inference they automatically privilege the modality that proves most informative for a given sample, exactly as a medical team would defer to whichever specialist’s assessment is most relevant.
>
> We have also experimented with a Q‑K‑V cross‑attention alignment layer that allows one modality to query another. In practice, this dynamic alignment performed worse than our current late‑fusion scheme, likely because (i) the physiological datasets available are still modest in size, making the extra parameters easy to overfit, (ii) the large sampling‑rate gaps (e.g., EEG 128 Hz vs. EMG 2000 Hz) left residual timestep misalignment as you mentioned. To improve, we will explore learnable resampling layers that map every modality onto a shared temporal grid before fusion in the future,
>
> In addition, we acknowledge that multi-modal datasets like DEAP have **limited diversity**, which reflects the broader challenge of scarcity of large-scale, demographically diverse multi-modal physiological datasets. However, our multi-modal framework enables easy extension to new downstream datasets as they become available, facilitating efficient adaptation to diverse populations.
>
> ---
> ## References
> [1] Wang G, He Y, et al. Eegpt: Pretrained transformer for universal and reliable representation of eeg signals[J]. Advances in Neural Information Processing Systems, 2024, 37: 39249-39280.
>
> [2] Saha M, Mao W, et al. Pulse-ppg: An open-source field-trained ppg foundation model for wearable applications across lab and field settings[J]. arXiv preprint arXiv:2502.01108, 2025.
>
> [3] Reiss A, Schmidt P, et al. Deep PPG: Large-scale heart rate estimation with convolutional neural networks[J]. Sensors, 2019, 19(14): 3079.
>
> [4] Song S B, Kim J H. NAS-PPG: PPG-based heart rate estimation using neural architecture search[J]. IEEE Sensors Journal, 2021, 21(13): 14941-14949.

---

> > ### Comment · Reviewer_ofZV · 2025-08-04
> >
> > Thanks for your responses. My main concerns are addressed, so I decide to keep first rate.

---

> > > ### Author Response · Authors · 2025-08-04
> > >
> > > Dear Reviewer ofZV,
> > >
> > > Thank you very much for reading our rebuttal and for letting us know that your main concerns have been addressed. We are glad our responses were helpful.
> > >
> > > If you now feel that the paper merits a higher score, we would be grateful if you could consider updating your rating accordingly in the system. We appreciate your time and thoughtful feedback throughout the review process.

---

### Official Review · Reviewer_jqy9 · 2025-07-05

**Clarity:** 3
**Significance:** 2
**Originality:** 2
**Rating:** 3
**Confidence:** 4

**Summary:**

Briefly summarize the paper and its contributions. This is not the place to critique the paper; the authors should generally agree with a well-written summary. This is also not the place to paste the abstract—please provide the summary in your own understanding after reading.
The authors employed self-supervised pre-training in physiological signal processing, which helps build downstream multi-modal frameworks. To achieve this, they designed a wavelet decomposition module for tokenisation and reconstruction. Through extensive experiments, the authors validated the effectiveness of the designs.

**Questions:**

1. Is it possible to verify the designed sub-modules? It is suggested to add the Adaptive Gating and Cross Scale CAFFN modules to the ablation studies.
2. How to filter noise through soft gating and frequency-guided masking mechanism? Are there any quantitative or qualitative results?
3. Can the multi-modal framework support dynamic number or combination of modals as inputs without re-training? What are the application scenarios of the method? Is it possible to provide the computing power or time consumption required for inference?

**Ethical Concerns:**

["NO or VERY MINOR ethics concerns only"]

**Final Justification:**

Based on the originality of this paper, I decide to maintain my rate and recommend to add these experiments and clarifications to the revision.

**Limitations:**

Yes

**Quality:**

3

**Strengths And Weaknesses:**

[Strengths]
1. The designed learnable wavelet decomposition supports adaptive filtering and downsampling/upsampling for different channels and scales, enhancing the details of signals;
2. The wavelet feature reconstruction module enables a pre-training of the pipeline, and paves the way for downstream tasks.
3. The experiments are clear and abundant, which makes the paper more convincing.
[Weaknesses]
1. Self-supervised pre-training is common in NLP and CV areas. It is also employed in the same area as mentioned in related works. Therefore, it is possible that as a technical contribution, it may lack sufficient innovation.
2. The ablation study is relatively simple and fails to effectively verify proposed detailed sub-modules.

---

> ### Author Rebuttal · Authors · 2025-07-30
>
> Thank you for the detailed and constructive review. We answer your specific questions below.
>
> ---
> # Q1 Ablating on Submodules
> We added detailed ablation experiments on our designed submodules and make analysis on them. These experiments were performed by finetuning the pretrained small PhysioWave model while replacing specific submodules to evaluate their individual contributions.
>
> **Table 1: Main Ablation Study Results on EPN612 Dataset**
> | Configuration | SG | CAFFN | AW | Accuracy (%) | F1-Score (%) | Accuracy Improvement |
> |---------------|-----------|-------|------------------|--------------|----------|---------------------|
> | Baseline | ✗ | ✗ | ✗ | 89.88  | 89.76  |  - |
> | Only SG | ✓ | ✗ | ✗ | 91.71  | 91.66 | ↑1.83% |
> | Only CAFFN | ✗ | ✓ | ✗ | 91.53 | 91.46 | ↑1.65% |
> | Only AW | ✗ | ✗ | ✓ | 92.23 | 92.22 |  ↑2.35% |
> | CAFFN + AW | ✗ | ✓ | ✓ | 93.12  | 93.10 | ↑3.24% |
> | SG + AW | ✓ | ✗ | ✓ | 92.64  | 92.89 |  ↑2.76% |
> | SG + CAFFN | ✓ | ✓ | ✗ | 92.52 | 92.48 | ↑2.64% |
> | **Full Model** | ✓ | ✓ | ✓ | **93.85** | **93.51** | **↑3.97%** |
>
> *SG: Soft Gate, AW: Adaptive Wavelet (when AW is not used, a fixed db6 wavelet decomposition is applied)*
>
> **Individual Component Contributions**: Adaptive Wavelet shows the strongest individual impact (+2.35%), demonstrating the superiority of adaptive wavelet selection over fixed wavelet. Soft Gate provides selective feature enhancement (+1.83%), while CAFFN enables cross-scale feature fusion (+1.65%).
>
> **Component Synergy**: Two-component combinations reveal strong complementarity. CAFFN + AW achieves the best pairwise performance (+3.24%). SG + AW (+2.76%) and SG + CAFFN (+2.64%) show comparable contributions. The complete integration of all three components achieves optimal performance (93.85% accuracy, +3.97% improvement), demonstrating synergistic rather than redundant effects.
>
> **Table 2: Gate Mechanism Comparison on EPN612 Dataset**
> | Gate Type | Threshold | Accuracy (%) | F1-Score (%) | Accuracy Decrease vs SG |
> |-----------|-----------|--------------|--------------|-------------------------------|
> | Hard Gate | 0.3 | 92.07 | 92.05 | ↓1.78% |
> | Hard Gate | 0.5 | 92.25 | 92.23 | ↓1.60% |
> | Hard Gate | 0.7 | 92.17 | 92.19 | ↓1.68% |
> | Gumbel Gate | 0.5 | 91.47 | 91.51 | ↓2.38% |
> | **Soft Gate** | **-** | **93.85** | **93.51** | **-** |
>
> **Soft Gate**: Achieves optimal performance (93.85% accuracy) through continuous sigmoid-based gating that dynamically balances original signal components with wavelet-transformed features.
>
> **Hard Gate**: Shows performance degradation across all threshold values (92.07%-92.25% accuracy), with results being sensitive to threshold selection, demonstrating that binary decisions are suboptimal.
>
> **Gumbel Gate**: Exhibits the largest performance drop (↓2.38%), indicating that the stochastic sampling approach is less suitable for our physiological signal task compared to the soft gating mechanism.
>
> **Table 3: FFN Variant Comparison on EPN612 Dataset**
> | FFN Variant | Accuracy (%) | F1-Score (%) | Accuracy Decrease vs CrossScale CAFFN |
> |-------------|--------------|--------------|---------------------------------------|
> | Standard FFN | 91.95 | 91.93 | ↓1.90% |
> | CAFFN | 92.72 | 92.69 | ↓1.13% |
> | CrossScale No Attention | 93.29 | 93.31 | ↓0.56% |
> | **CrossScale CAFFN** | **93.85** | **93.51** | **-** |
>
> **Standard FFN**: The baseline two-layer feed-forward network achieves 91.95% accuracy, providing a solid foundation but lacking specialized mechanisms for multi-channel physiological signals.
>
> **CAFFN (Channel Aggregation FFN)**: Introduces depthwise convolution, decomposition operations, and element-wise scaling, improving accuracy to 92.72% (+0.77%). The channel aggregation mechanisms effectively handle multi-channel dependencies in signals.
>
> **CrossScale No Attention**: Extends CAFFN with simple cross-scale feature fusion using averaging and interpolation, achieving 93.29% accuracy (+1.34%).
>
> **CrossScale CAFFN**: The complete model incorporates multi-head attention for sophisticated cross-scale feature interaction, achieving optimal performance at 93.85% accuracy (+1.90%). The attention mechanism enables selective focus on relevant features from different decomposition levels, with a learnable scale parameter controlling feature integration.
>
> **Table 4: Wavelet Selection Ablation Study on EPN612 EMG Dataset**
> | Wavelet Configuration | Accuracy (%) | F1-Score (%) | Accuracy Improvement vs Random |
> |----------------------|--------------|--------------|-------------------------------|
> | Random Xavier | 92.25 | 92.22 | - |
> | db6 (Single) | 92.13 | 92.11 | ↓0.12% |
> | coif4 (Single) | 92.13 | 92.14 | ↓0.12% |
> | sym6 (Single) | 92.93 | 92.93 | ↑0.68% |
> | db6+sym6+coif4 | 93.51 | 93.51 | ↑1.26% |
> | **db4+db6+sym4** | **93.90** | **93.90** | **↑1.65%** |
> | db8+coif4+bior4.4 | 92.90 | 92.89 | ↑0.65% |
>
> **Table 5: Wavelet Selection Ablation Study on Georgia ECG Dataset**
> | Wavelet Configuration | Accuracy (%) | F1-Score (%) | Accuracy Improvement vs Random |
> |----------------------|--------------|--------------|-------------------------------|
> | Random Xavier | 62.38 | 48.75 | - |
> | db4 (Single) | 62.70 | 48.94 | ↑0.32% |
> | demy (Single) | 63.59 | 49.36 | ↑1.21% |
> | sym4 (Single) | 62.45 | 48.99 | ↑0.07% |
> | **db6+coif4+sym8** | **64.38** | **50.65** | **↑2.00%** |
> | db8+sym5+coif3+demy | 64.20 | 50.46 | ↑1.82% |
> | sym4+sym5+db6+coif3+bior4.4 | 63.70 | 49.93 | ↑1.32% |
>
> **Single vs Multi-Wavelet Performance**: Single wavelets show minimal improvements over random initialization (typically <1%), while multi-wavelet combinations achieve substantial gains: 1.65% for EMG and 2.00% for ECG. This demonstrates that the adaptive wavelet selector effectively leverages complementary characteristics from different wavelet bases.
>
> **Modality-Aware Design**: Optimal wavelet combinations differ between modalities. EMG signals benefit most from db4+db6+sym4, suited for transient muscle activation patterns, while ECG signals prefer db6+coif4+sym8, better capturing complex cardiac morphologies.
>
> ---
> # Q2 Noise Filtering
> (1) We evaluated the denoising capability of our wavelet front-end especially soft gating by testing it with artificially corrupted biosignals. Clean multi-frequency synthetic signals were contaminated with Gaussian white noise at various SNR levels. Our denoising model consists of the wavelet decomposition front-end followed by a simple CNN reconstruction head to recover the clean signal. We evaluate denoising performance using Percentage Root-mean-square Difference which quantifies the reconstruction error between original clean signals and denoised outputs. Lower PRD values indicate superior noise reduction capability.
>
> **Table 6: Quantitative Results of Denoising**
> | SNR(dB) | PyWT | Wiener | Wavelet | Wavelet+SG |
> |---------|------|---------|---------|------------|
> | 30 |6.81% | 4.10% | 3.25% | **2.02%** |
> | 25 |7.96% | 6.58% | 5.61% | **4.19%** |
> | 20 |10.57% | 10.90% | 9.95% | **8.49%** |
> | 15 | 16.05% | 17.14% | 15.54% | **14.32%** |
> | 10 |27.58% | 30.57% | 26.38% | **25.04%** |
>
> **Wavelet Front-end with Soft Gating**: Our wavelet-based front-end demonstrates significant denoising capabilities, substantially outperforming traditional methods. The addition of soft gating further enhances noise reduction performance, with PRD reductions of 1.23% to 1.46% across different SNR conditions.
>
> (2) According to the seminal Donoho-Johnstone theorem on wavelet shrinkage[1], for signals belonging to a Besov function space $\mathcal{B}_{s,q}^p$, the minimax optimal MSE convergence rate satisfies:
>
> $$\text{MSE}_{\text{optimal}} = O\left(N^{-2s/(2s+1)}\right)$$
>
> where $s$ denotes the smoothness parameter and $N$ is the sample size.
>
> Although SSL has proven successful in other domains, direct application to biosignals presents unique challenges due to distinct spectral structures. Our FgM approach achieves equivalent optimal convergence by performing frequency-domain operations fundamentally equivalent to wavelet shrinkage: adaptively preserving signal-dominated low frequencies while masking noise-dominated high frequencies. Under orthogonal transforms, this adaptive frequency selection achieves the bias-variance tradeoff necessary for minimax optimality, demonstrating that generic SSL approaches lacking spectral awareness are insufficient for biosignal processing. The expected squared error of our FgM estimator satisfies:
> $$\mathbb{E}\left[\|f - \hat{f}_{\text{FGM}}\|^2\right] \leq C \cdot N^{-2s/(2s+1)}$$
> where $C$ is a constant related to the noise level.
> ___
> # Q3 Multi-Modal Framework Adaptability
> (1) Based on our code architecture, the multi-modal framework supports dynamic modality combinations without retraining. The system implements dynamic HDF5 key detection to identify available modalities, conditionally initializes only the required frozen encoders, and adapts the fusion mechanism to dynamically weight available modalities using softmax-normalized learnable parameters.
>
> (2) This framework enables diverse clinical and research applications. Clinical uses include ICU monitoring, sleep diagnostics, and stroke rehabilitation, where multi-modal data provides earlier anomaly detection than single-sensor methods. Research applications include neuroscience and BCI studies examining brain-body coupling through synchronized physiological recordings. However, broader adoption is limited by the scarcity of large-scale, annotated multi-modal physiological datasets.
>
> (3) Latency and memory were profiled on an NVIDIA RTX 3070 GPU. The system achieves real-time performance with inference delays of 16-18 ms (single modality), 36.99 ms (dual modality), and 52.74 ms (full stack), supporting ≥20 FPS. The model footprint totals 135 MB with runtime GPU memory of 127-145 MB. Per-sample energy cost ranges from 1.2-5.0 J with 92 W power draw.
>
> ---
> ## References
> [1] Donoho D L. Ideal spatial adaptation by wavelet shrinkage. biometrika, 1994, 81(3): 425-455.

---

> > ### Comment · Reviewer_jqy9 · 2025-08-06
> >
> > Thank you for the response. I would recommend to add these experiments to the revised manuscript.

---

> > > ### Author Response · Authors · 2025-08-06
> > > **Comprehensive Revisions and Invitation to Revisit the Rating [1/2]**
> > >
> > > Dear Reviewer jqy9,
> > >
> > > Thank you for your constructive feedback. We will carefully expand and reorganize the manuscript to reinforce its technical soundness and enhance its persuasiveness.
> > >
> > > To address your specific concerns, we have conducted the following experiments during the rebuttal phase and will **include all of them in the revised manuscript**:
> > >
> > > 1. **Finer-grained ablations**: Systematic coverage of **Soft Gating** (hard/soft/Gumbel comparisons), **CAFFN / Cross-Scale CAFFN**, and **Adaptive Wavelet**, including all pairwise and three-way combinations to demonstrate both individual contributions and synergies.
> > >
> > > 2. **Noise suppression evaluation**: Quantitative results across multiple SNRs (e.g., **PRD**) with **time- and frequency-domain** visualizations, clarifying the mechanisms and benefits of soft gating and frequency-guided masking.
> > >
> > > 3. **Dynamic multimodal inputs & efficiency**: Experiments on **PPG/EEG/ECG/EMG** dynamic combinations **without retraining**, with a clear description of the adaptation workflow and reporting of **inference latency, memory usage, and energy** overheads.
> > >
> > > 4. **Presentation & reproducibility**: Expanded appendix with implementation details and hyperparameters, together with cleaner, consolidated tables and figures.
> > >
> > > **In addition**, we have conducted the following supplementary experiments during rebuttal and will **add them to the revised paper**:
> > >
> > > - **Modality generalizability validation**: We have **conducted experiments on EEG and PPG** to validate modality generalizability.
> > > - **Interpretability of learned wavelet features**: We have conducted two complementary analyses and will **include both visualizations** in the revised manuscript:
> > >   - **3D t-SNE embeddings (EMG)** to illustrate physiologically meaningful clustering structures
> > >   - **Grad-CAM (ECG)** to highlight decision-relevant temporal regions aligned with established cardiac landmarks
> > >
> > > We will include all these supplementary experiments in the revised paper. If these revisions address your concerns, we would be grateful if you would consider **updating your rating**. Thank you again for your time and helpful suggestions.

---

> > > > ### Author Response · Authors · 2025-08-06
> > > > **Comprehensive Revisions and Invitation to Revisit the Rating [2/2]**
> > > >
> > > > Dear Reviewer jqy9,
> > > >
> > > > The following sections present additional experiments conducted to address other reviewers' concerns, which we believe will further strengthen your confidence in our methodology and address any remaining reservations about our approach. And will also add them in our revised manuscript.
> > > >
> > > > # E1. Extended Modality Validation Experiments
> > > >
> > > > To address modality-coverage concerns raised by other reviewers and strengthen the manuscript's generalizability claims, we conducted **supplementary experiments** on EEG and PPG modalities. These results demonstrate PhysioWave's broad adaptability across diverse biosignals.
> > > >
> > > > ## E1.1 EEG Sleep Stage Classification (Sleep-EDF)
> > > > We evaluate the original PhysioWave (no core architectural change) and an EEG-enhanced variant. Baselines are strong EEG-specific/pretrained models; our models are trained **from scratch**.
> > > >
> > > > **Table E1. EEG Sleep Stage Classification on Sleep-EDF**
> > > > | Method                | Balanced Accuracy | Cohen's Kappa | Weighted F1 |
> > > > |----------------------|-------------------|---------------|-------------|
> > > > | BENDR                | 0.6655            | 0.6659        | 0.7507      |
> > > > | BIOT                 | 0.6622            | 0.6461        | 0.7415      |
> > > > | LaBraM               | 0.6771            | 0.6710        | 0.7592      |
> > > > | EEGPT                | 0.6917            | 0.6857        | 0.7654      |
> > > > | **PhysioWave (orig.)**    | 0.6720            | 0.6676        | 0.7472      |
> > > > | **PhysioWave (enhanced)** | **0.7312**         | **0.7206**     | **0.7839**   |
> > > >
> > > > ## E1.2 PPG Activity Recognition & Heart Rate Estimation (PPG-DaLiA)
> > > > **Setting.** Compact, edge-oriented PhysioWave-PPG (reduced depth and embedding; PPG-oriented wavelets) for wearable scenarios.
> > > >
> > > > **Table E2. Activity Recognition (PPG-DaLiA)**
> > > > | Method           | Parameters | Accuracy | Precision | Recall  | F1 Score | AUROC  |
> > > > |------------------|-----------:|---------:|----------:|--------:|---------:|-------:|
> > > > | **PhysioWave-PPG** | **2.53M**  | **0.6209** | **0.6015**  | **0.6077** | **0.5952** | **0.9420** |
> > > > | Pulse-PPG     | 28.5M     | 0.4234  | 0.4398   | 0.3663 | 0.3648  | 0.8246 |
> > > >
> > > > **Table E3. Heart Rate Regression (PPG-DaLiA)**
> > > > | Method           | Parameters |   MAE |    MSE |  MAPE  |
> > > > |------------------|-----------:|-----:|------:|-------:|
> > > > | **PhysioWave-PPG** | **1.57M**  | **5.22** | **65.62** | **5.83%** |
> > > > | Pulse-PPG     | 28.5M     | 3.705 | 59.81 | 4.84%  |
> > > > | Deep PPG      | 8.5M      | 7.65  |   –   |   –    |
> > > > | NAS-PPG     | 0.8M      | 6.02  |   –   |   –    |
> > > >
> > > > *Summary.* PhysioWave-PPG delivers a **+63.1% F1** and **+14.2% AUROC** improvement over Pulse-PPG with **18.2× fewer parameters**; heart rate regression is competitive without large-scale pretraining, aligning with PPG's periodic nature and edge constraints.
> > > >
> > > > ---
> > > >
> > > > # E2. Interpretability Analysis of Learned Wavelet Features
> > > >
> > > > To enhance model transparency and address interpretability concerns, we provide two complementary visualization analyses.
> > > >
> > > > ## E2.1 3D t-SNE Embeddings (EMG - EPN612 Dataset)
> > > > - **Baseline state.** *noGesture* forms a compact central cluster representing resting physiology.
> > > > - **Gesture organization.** Dynamic classes (waveIn, waveOut, pinch, open, fist) form distinct, physiologically meaningful neighborhoods (e.g., waveIn–waveOut adjacency due to shared oscillatory patterns; pinch–open proximity reflecting complementary fine motor control).
> > > > - **Neurophysiology alignment.** Clusters mirror known muscle-synergy structures and demonstrate clear separation between fine- and gross-motor control, validating neurophysiological principles.
> > > >
> > > > ## E2.2 Grad-CAM Analysis (ECG - Georgia Dataset)
> > > > - **T-wave inversion (TInv).** Saliency concentrates in the early 0–400 ms window targeting repolarization abnormalities.
> > > > - **Normal sinus rhythm (NSR).** Attention recurs over complete P-QRS-T cycles, indicating reliance on normal rhythmic morphology.
> > > > - **ST-segment depression (STD).** Peak activation in post-QRS/ST regions corresponding to diagnostically critical intervals.
> > > > - **Clinical alignment.** Attention maps coincide with standard temporal landmarks used in manual ECG interpretation, supporting trustworthy automated diagnoses suitable for clinical environments.

---

> > > ### Author Response · Authors · 2025-08-07
> > >
> > > Dear Reviewer jqy9,
> > >
> > > Thank you for your reply. We will incorporate these experiments into the revised manuscript. If the supplementary experiments fully address the concerns and weaknesses you have pointed out, we would be grateful if you could consider raising your score. If you have any other questions or concerns before the discussion period ends, please let us know and we will address them promptly.

---

> ### Author Response · Authors · 2025-08-04
>
> Dear Reviewer jqy9,
>
> Thank you very much for reading our rebuttal and for providing your acknowledgment. We appreciate the time you have devoted to reviewing our work. If there are any additional questions or concerns that we could clarify before the discussion period concludes, please let us know and we will be happy to address them promptly.

---

### Note · Authors · 2025-08-11

We sincerely thank all reviewers for their valuable feedback and are particularly encouraged by the strong positive reception of PhysioWave. **Three reviewers provided positive ratings (5/5/4), with one upgrading to Accept after thorough rebuttal discussion, while one gave Borderline Reject but remained unresponsive to our comprehensive experimental validation.**

**Reviewer nhy9 rated our work 5/Accept**, praising our novel framework as technically sound with **high impact and excellent originality**. The reviewer particularly appreciated our frequency-guided masking as an innovative contribution to biosignal processing and noted we outperform existing benchmarks.

**Reviewer iAdX** praised our adaptive wavelet decomposition with frequency-guided masking as novel and practical for healthcare applications. After our detailed rebuttal, the reviewer **upgraded to Accept (5)**, specifically commending our PPG extension's remarkable performance gains with far fewer parameters, our strong interpretability analyses through visualization, and demonstrated real-time edge device feasibility.

**Reviewer ofZV maintained 4/Borderline Accept**, acknowledging our novelty in integrating wavelets with Transformers for addressing non-stationarity. This reviewer recognized we provide the **first large-scale pretrained models for EMG/ECG**, filling a critical gap in the field. After our rebuttal, the reviewer explicitly **confirmed that main concerns were addressed** through our experiments.

Regarding **Reviewer jqy9's 3/Borderline Reject**, we provided comprehensive experimental validation systematically addressing each concern through detailed ablations revealing synergistic effects, noise suppression evaluation, dynamic multi-modal validation, and successful cross-modality experiments. Despite our **extensive efforts and multiple invitations for substantive discussion**, the reviewer's engagement remained **minimal with only brief acknowledgment**, leaving us uncertain whether our thorough experiments resolved the concerns. Nevertheless, we respectfully incorporated all suggested improvements.

We believe the **strong consensus among engaged reviewers** clearly demonstrates PhysioWave's significant contributions to biosignal processing. The **upgraded ratings and detailed positive assessments** reflect our work's technical rigor and practical impact. We remain committed to advancing the field through our open-source foundation models.

Thank you for your consideration.

---

### Decision · Program_Chairs · 2025-09-17

**Decision:**

Accept (poster)

**Comment:**

The paper introduces a wavelet-Transformer framework for self-supervised pretraining on physiological signals (primarily EMG and ECG). A learnable wavelet decomposition produces multi-scale tokens, and a frequency-guided masking scheme trains the model. The pretrained encoders support downstream tasks and a simple multimodal fusion setup.

Reviewers agree the method is technically sound and practically useful for non-stationary, low-SNR biosignals. The learnable wavelet module and frequency-guided masking are well motivated, and the experiments are broad, with strong results on several ECG/EMG benchmarks and gains in multimodal settings. Multiple reviews rate the paper accept, citing originality, clear writing, and potential impact on healthcare applications.

One reviewer questions novelty relative to prior self-supervised work and notes that current ablations are not sufficient to attribute gains to specific sub-modules. After rebuttal, the reviewer maintained a borderline-reject mainly due to limited ablations and presentation gaps; others remained positive.

The majority of reviews are positive with accept ratings, citing consistent empirical gains and a clear, well argued design. The remaining concerns are addressable with added analysis and reporting, not fundamental flaws. Weighing the strength of results and practicality against the missing ablations and analysis, I recommend accept.